# Comparative therapeutic strategies for preventing aortic rupture in a mouse model of vascular Ehlers-Danlos syndrome

**Anne Legrand**[1,2], **Charline Guery**[1], **Julie Faugeroux**[1], **Erika Fontaine**[1], **Carole Beugnon**[1], **Amélie Gianfermi**[1], **Irmine Loisel-Ferreira**[1], **Marie-Christine Verpont**[3], **Salma Adham**[2], **Tristan Mirault**[1,2,4], **Juliette Hadchouel**[5], **Xavier Jeunemaitre**[1,2] *

**1** Université de Paris, INSERM, U970, Paris Cardiovascular Research Centre, Paris, France, **2** Assistance Publique-Hôpitaux de Paris, Hôpital Européen Georges Pompidou, Service de Génétique et Centre de Référence des Maladies Vasculaires Rares, Paris, France, **3** Sorbonne Université, INSERM, U1155, Plateforme d'Imagerie et de Cytométrie, Paris, France, **4** Assistance Publique–Hôpitaux de Paris, Hôpital Européen Georges-Pompidou, Service de Médecine Vasculaire, Paris, France, **5** Sorbonne Université, INSERM, U1155, Hôpital Tenon, Paris, France

* xavier.jeunemaitre@inserm.fr

**Data Availability Statement:** RNA sequencing data are available from the Gene Expression Omnibus (GEO) under accession GSE196753. All other

## Abstract

Vascular Ehlers-Danlos syndrome is a rare inherited disorder caused by genetic variants in type III collagen. Its prognosis is especially hampered by unpredictable arterial ruptures and there is no therapeutic consensus. We created a knock-in Col3a1[+/G182R] mouse model and performed a complete genetic, molecular and biochemical characterization. Several therapeutic strategies were also tested. Col3a1[+/G182R] mice showed a spontaneous mortality caused by thoracic aortic rupture that recapitulates the vascular Ehlers-Danlos syndrome with a lower survival rate in males, thin non-inflammatory arteries and an altered arterial collagen. Transcriptomic analysis of aortas showed upregulation of genes related to inflammation and cell stress response. Compared to water, survival rate of Col3a1[+/G182R] mice was not affected by beta-blockers (propranolol or celiprolol). Two other vasodilating anti-hypertensive agents (hydralazine, amlodipine) gave opposite results on aortic rupture and mortality rate. There was a spectacular beneficial effect of losartan, reversed by the cessation of its administration, and a marked deleterious effect of exogenous angiotensin II. These results suggest that blockade of the renin angiotensin system should be tested as a first-line medical therapy in patients with vascular Ehlers-Danlos syndrome.

## Author summary

Vascular Ehlers-Danlos syndrome (vEDS) is a rare vascular genetic disease leading to life-threatening arterial and colonic fragility in young adulthood. We created a new mutant mouse with a typical disease-causing variant in the gene responsible for vEDS. This mouse recapitulates the vEDS vascular features with spontaneous mortality due to aortic rupture. We also tested several antihypertensive therapeutic strategies to improve the survival of this mouse. Only one of the 5 tested medications, losartan, which blocks the

relevant data associated with this study are within the manuscript and its Supporting Information files.

**Funding:** This work was supported by INSERM (www.inserm.fr); the Fondation pour la Recherche Médicale (https://www.frm.org/), grant "Equipes FRM" 2015, grant number DEQ20150331716; the Agence Nationale pour la Recherche (https://anr.fr/), grant number ANR-14-CE15-0012-02; the Association Française pour les Syndromes d'Ehlers Danlos (AFSED, www.afsed.com) and the Association David (https://associationdavid.org/). A.L. has obtained a 3-year Ph.D grant from Région Ile-de-France and a one-year PhD grant from the Fondation pour la Recherche Médicale (FRM), grant number FDT202001010883. The funders had no role in study design, data collection and analysis, decision to publish, or preparation of the manuscript.

**Competing interests:** The authors have declared that no competing interests exist.

activity of angiotensin II, a vasoconstricting hormone, improves the survival of this mouse. Moreover, the deleterious effect of angiotensin II administration further highlights the role of angiotensin II on susceptibility to aortic rupture in this mouse. These results support the interest of a therapeutic trial in vEDS patients using angiotensin II receptor blockers.

## Introduction

Vascular Ehlers-Danlos syndrome (vEDS, OMIM #130050) is a rare autosomal dominant inherited condition caused by genetic defects in the *COL3A1* gene coding for collagen III (OMIM *120180). Collagen III is one of the major fibrillary collagens, especially found in the arterial wall, the skin and hollow organs such as the uterus and colon [1]. The severity of vEDS is mainly explained by the occurrence of arterial dissections and ruptures as well as colonic perforations in young adulthood, reducing dramatically the life-expectancy [2]. In addition, pregnancies are characterized by prematurity and increased maternal morbidity [3]. Two types of genetic variants are causing the disease [4,5]. The most frequent type corresponds to dominant-negative variants and represents roughly 80% of symptomatic cases. They are either missense variants occurring at one of the 343 glycine residues of the triple helical domain of collagen III (https://eds.gene.le.ac.uk/home.php?select_db=COL3A1) (65% of the cases) or splice site variants leading to in-phase exon skipping (15% of the cases). Since three identical proα1(III) chains are wrapped together into the triple helix to form an homotrimer, both types of heterozygous (htz) variants result in a dominant negative effect and are responsible for the most severe vEDS cases [4,5]. Less frequently, truncating variants leading to haploinsufficiency, are responsible for a milder phenotype [4–6].

The main role of collagen in arteries is the maintenance of arterial structure and resistance to arterial blood pressure. Indeed collagen fibers are stretched at higher than normal pressures and have a protective supporting role, whereas elastic fibers characterized by their great range of extensibility limit the normal blood pressure (BP) fluctuations [7]. In vEDS patients, the marked arterial fragility caused by the deficit in collagen III is associated with thinner arterial wall and slightly increased stiffness at diastolic blood pressure [8,9]. When BP increases during physiological or pathological circumstances, the defective stiffening facilitates arterial rupture [9]. Thus, even though vEDS patients are almost always normotensive at basal state, these anomalies suggest a possible beneficial effect of BP lowering drugs [9]. The BBEST study, the only randomized open clinical trial performed to date in this rare disease demonstrated that celiprolol, a β-blocker with β1 antagonist and β2 agonist activities, could achieve a reduction in arterial morbidity and mortality [10]. More recently, the beneficial effect of this drug was also suggested in a long-term observational study with a better survival rate in patients treated with celiprolol [11]. The possible efficacy of other BP lowering drugs for reducing arterial events has not been tested in humans and is difficult to set up in a double blind randomized controlled manner because of the rarity of vEDS and of ethical issues of such trials in severe and rare conditions [2].

In the face of such difficulties, animal experimental models can be particular useful not only to explore unknown biological process but also new therapeutic schemes. Liu et al. [12] created a first constitutive *Col3a1* knock-out mouse model. Most of homozygous mutants died *in utero* or survived only few days after birth, making it unusable for physiological and pharmacological studies. In this model, htz *Col3a1*$^{+/-}$ deficient mice expressed no spontaneous vascular phenotype but only late-onset histological vascular lesions [13]. Our team showed the

high sensitivity of this model to angiotensin II (Ang II) with thoracic aorta ruptures occurring within days after starting the infusion [14]. Another mouse model examined by Smith et al. [15] was initially thought to be haploinsufficient for *Col3a1*, but recent transcript analysis revealed that the 185 kb deletion led to only exons 33–39 deletion and a dominant negative mechanism [16]. Just over one quarter of htz mutant died early from aortic rupture [15]. *Ex vivo*, the aortic wall integrity in this model was improved by celiprolol whereas bisoprolol and losartan had no effect [16,17]. Another transgenic mice overexpressing *Col3a1* with a glycine substitution (*Col3a1^Tg-G182S^*) showed an early death at 13–14 weeks due to severe skin wounds before presenting spontaneous arterial phenotype [18]. Recently, two knock-in models *Col3a1^G209S/+^* and *Col3a1^G938D/+^* recapitulated vascular vEDS phenotype with spontaneous mortality caused by aortic rupture. Enhanced PLC/IP$_3$/PKC/ERK pathway was proposed to explain the pathophysiology, this hypothesis being strengthened by the efficiency of hydralazine [19] on short-term mice survival.

We herein report the creation of a new knock-in *Col3a1* mouse model affecting a glycine residue of the triple helix. *Col3a1^+/G182R^* mice recapitulate most of the features of vEDS patients and show a high frequency of spontaneous aortic rupture in the first 24 weeks. Pharmacological challenges suggest the high possible benefit of a treatment based on Ang II receptor blockade in this pathology.

## Results

### Survival follow-up of the *Col3a1^+/G182R^* model

To study the consequences of mutant type III collagen, we generated a knock-in *Col3a1* mouse on a C57BL/6J background, using bacterial artificial chromosome (BAC) and recombinant technology (for details, see Materials and Methods section, S1 Fig and S1 Table). We chose the most common type of *COL3A1* disease-causing genetic variant, i.e. a glycine substitution within the triple helix of the mature type III collagen, and one (p.G182R) already observed in vEDS patients (https://eds.gene.le.ac.uk/home.php?select_db=COL3A1). The htz mice will be referred to as *Col3a1^+/G182R^* and wild-type (WT) mice as *Col3a1^+/+^* throughout the manuscript.

Once mice carrying the *Col3a1* p.G182R allele were obtained, we conducted breeding between *Col3a1^+/G182R^* mice. Whereas *Col3a1^+/G182R^* mice were perfectly viable, a significant deficit of homozygous *Col3a1^G182R/G182R^* mice was observed at the 4 weeks genotyping (9% vs. 25% expected, p<10$^{-4}$). The mortality rate was evaluated after weaning, around 3–4 weeks of age, up to 24 weeks of age. The homozygous *Col3a1^G182R/G182R^* mice had a strong mortality rate compared to *Col3a1^+/G182R^* mice (80.0% vs. 35.5%, p = 0.001, Fig 1A) at 24 weeks of age. For the few animals that we were able to analyze, they had lower body weight (14.2 ± 1.1 g, p = 0.049; n = 10) and lower systolic blood pressure (SBP) (103.0 ± 7.3 mmHg; p = 0.062; n = 4) at 5 and 8 weeks of age, respectively.

Therefore, we mainly characterized the *Col3a1^+/G182R^* mice by breeding *Col3a1^+/G182R^* with *Col3a1^+/+^* mice. *Col3a1^+/G182R^* mice showed an important rate of spontaneous mortality compared to *Col3a1^+/+^* littermates (35.5% vs. 0.0%, p<10$^{-4}$, Fig 1A). Interestingly, this mortality was higher in males than females out of breeding, (56.1% vs. 11.1%, p<10$^{-4}$, Fig 1B). For breeding females significant mortality was noticed during the lactation periods with a 54.5% (n = 6/11) mortality rate after the first lactation, up to 81.8% (n = 9/11) after the third one (S2 Fig). Of the 9 observed deaths, two involved both lactating and pregnant mice (during the second lactation) and the other 7 involved lactating mice only.

Globally, the SBP was lower in *Col3a1^+/G182R^* with 110.2 ± 2.1 vs. 118.2 ± 2.2 mmHg, (p = 0.012, Fig 1C and S2 Table) at 24 weeks of age in surviving mice. Of note, *Col3a1^+/G182R^*

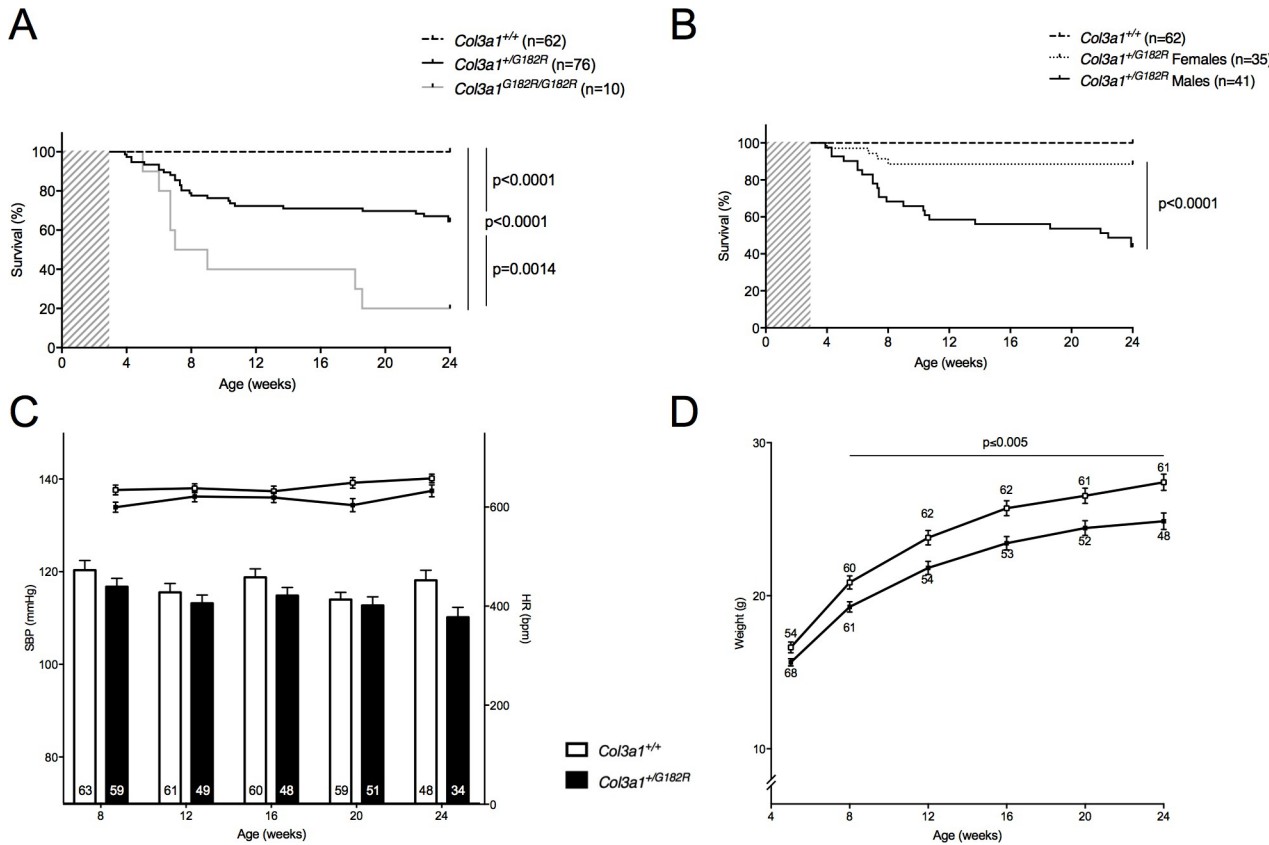

**Fig 1. Survival rate and basal hemodynamic parameters of *Col3a1*$^{+/G182R}$ mice. A** Kaplan-Meier Survival curve for comparing *col3a1*$^{+/+}$ (n = 62) to *Col3a1*$^{+/G182R}$ (n = 76) and to *Col3a1*$^{G182R/G182R}$ (n = 10), which die from vascular rupture or dissection. Significant differences are calculated using Log-Rank (Mantel-Cox) analysis. **B** Kaplan-Meier Survival curve for comparing *Col3a1*$^{+/G182R}$ males (n = 41) and *Col3a1*$^{+/G182R}$ females (n = 35). Significant difference is calculated using Log-Rank (Mantel-Cox) analysis. **C** Graph represents basal systolic blood pressure (SBP; lower bars: left scale) and heart rate (HR; upper lines: right scale), measured between 8 and 24 weeks in the same *Col3a1*$^{+/+}$ and *Col3a1*$^{+/G182R}$ mice represented in A using a tail-cuff method. Numbers within the bars indicate the number of living mice studied at each time of measurement. SBP was comparable between groups of mice (p>0.05 at each time of measurement: 8, 12, 16 and 20, except for 24 weeks, student t-test) and HR was lower in *Col3a1*$^{+/G182R}$ (p<0.05 at 8 and 20 weeks, student t-test). **D** Graph represents weight measured between 5 and 24 weeks in the same *Col3a1*$^{+/+}$ and *Col3a1*$^{+/G182R}$ mice represented in A. Weight was significantly different in *Col3a1*$^{+/+}$ and *Col3a1*$^{+/G182R}$ mice (p = 0.020 at 5 weeks and p<0.005 between 8 and 24 weeks, student t-test). Data are expressed as the mean ± SEM.

mice were 10% lighter (24.9 ± 0.5 vs. 27.4 ± 0.5 g; p = 0.001, Fig 1D and S2 Table), reflecting what is generally observed in vEDS patients [5]. This difference was observed in both males and females (S2 Table).

## Early spontaneous mortality of *Col3a1*$^{+/G182R}$ untreated mice caused by aortic rupture

Dead mice were autopsied. Hemothorax was observed in almost all cases with hematomas surrounding the heart, occurring spontaneously in males or during gestation or lactation in most of the females (Fig 2A and 2B). Macroscopic analysis confirmed that all deaths were due to aortic ruptures occurring at the level of the thoracic aorta in 93.8% (n = 45/48) (Fig 2C). To assess if any arterial dilation preceded the rupture, aortic echography was performed in *Col3a1*$^{+/+}$ (n = 7) and *Col3a1*$^{+/G182R}$ (n = 4) male mice at 6 weeks of age showing no dilation (S3 Fig).

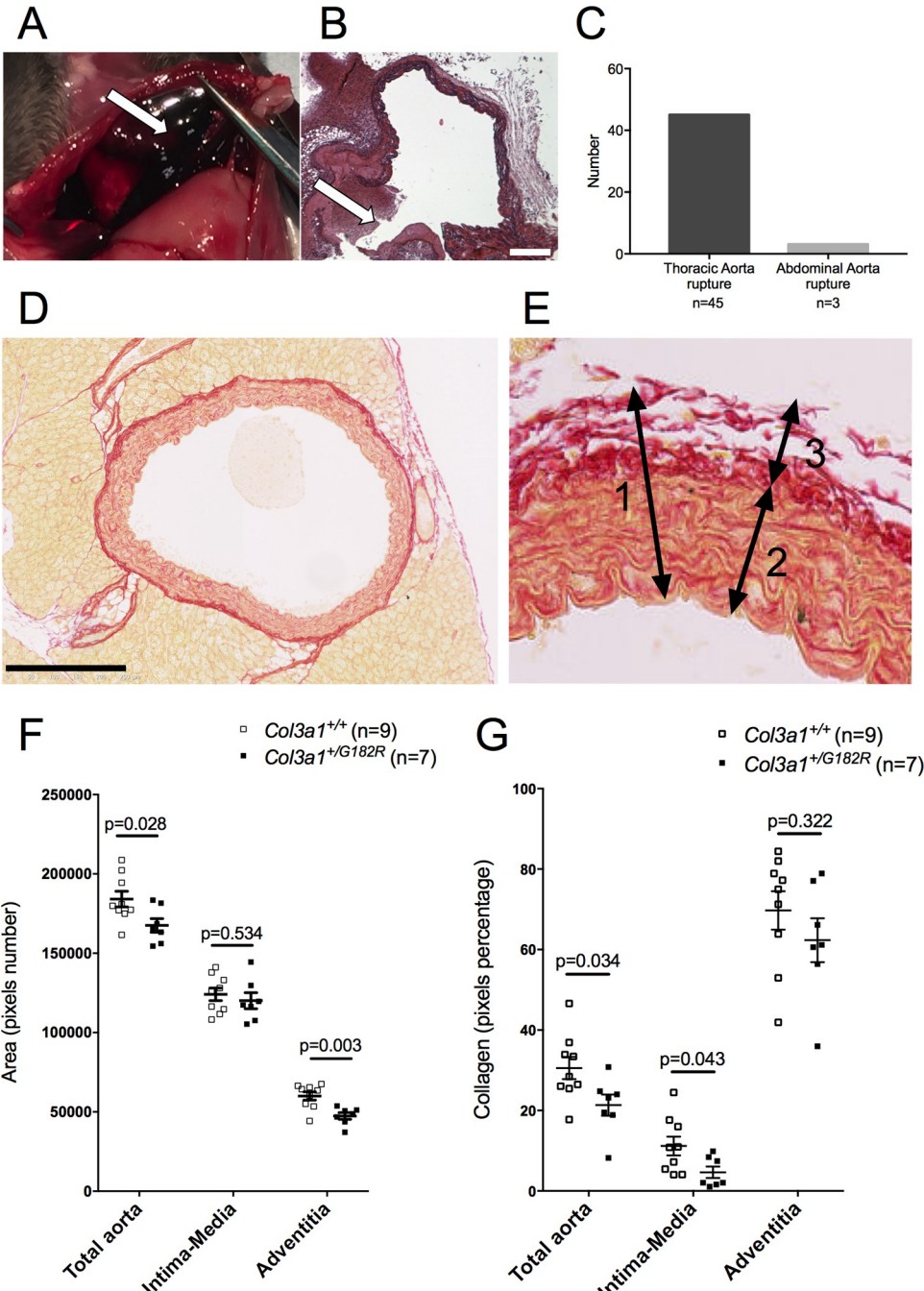

**Fig 2. Assessment of collagen content in *Col3a1^{+/G182R}* untreated mice (Picrosirius red staining).** **A** Hemothorax in *Col3a1^{+/G182R}* mice, white arrow shows the massive hemorrhage in the chest. **B** Histological staining (Hematoxylin) of the descending thoracic aorta of *Col3a1^{+/G182R}* dead mice. White arrow shows the spontaneous aortic rupture and hemorrhage. Scale bar = 50 μm. **C** Localization of the aortic dissection/rupture determined through dissection in *Col3a1^{G182R/+}* mice. **D** Histological staining (Picrosirius Red) of a descending thoracic aortic section in *Col3a1^{+/G182R}* mice. Scale bar = 250 μm. **E** details of a descending thoracic aortic section to show the two considered layers in the aortic wall (1): the intima-media (2) and the adventitia (3). The area and the collagen density in the adventitia were calculated by subtracting the intima-media pixels from the total aorta pixels. **F** The graph represents the area of the total aorta, the intima-media and the adventitia in *Col3a1^{+/+}* (n = 9) and *Col3a1^{+/G182R}* (n = 7) untreated male mice. The area of descending thoracic aorta and corresponding adventitia are lower in *Col3a1^{+/G182R}* mice compared to *Col3a1^{+/+}* mice (Student-t test, p<0.05). **G** The graph represents the collagen density in the total aorta, the intima-media and the adventitia in *Col3a1^{+/+}* (n = 9) and *Col3a1^{+/G182R}* (n = 7) untreated male mice. The collagen density in the total aorta and the intima-media are lower in *Col3a1^{+/G182R}* mice compared to *Col3a1^{+/+}* mice (Student-t test, p<0.05) whereas it remains constant in the adventitia (Student-t test, p>0.05). Data are expressed as the mean ± SEM.

## Abnormal collagen fibers and alteration of aortic adventitial fibroblast morphology

To assess if histological findings were associated with the aortic fragility, thoracic aortas from *Col3a1*$^{+/+}$ (n = 9) and *Col3a1*$^{+/G182R}$ (n = 7) 24-week-old surviving and untreated male mice were harvested and stained with Picrosirius red. The area of the aortas was significantly lower in *Col3a1*$^{+/G182R}$ mice mainly due to a thinner adventitia (area = 60.031 ± 2.565 vs 47.442 ± 2.106 pixels, n = 9 vs. n = 7, p = 0.003) and no change of the intima-media layer (Fig 2F). Density of the collagen fibers was reduced in the aortic wall of the *Col3a1*$^{+/G182R}$ mice (n = 9 vs. n = 7, p = 0.034), mainly in the intima-media (Fig 2G).

Then, electron microscopy was performed on thoracic aortas from *Col3a1*$^{+/+}$ (n = 3) and *Col3a1*$^{+/G182R}$ (n = 3) 24-week-old surviving and untreated mice. Analysis of collagen fibers showed abnormal heterogeneity of fiber diameters associated with a lower density (Figs 3F and S4). An abnormal morphology of adventitial fibroblasts with large vesicles surrounded by black particles (ribosomes) corresponding to dilated endoplasmic reticulum (ER) was observed (Fig 3D) but no anomaly of the vascular smooth muscle cells (vSMC), elastic laminae or endothelial cells (Fig 3B).

We also investigated whether the observed ER increased volume was associated with ER stress and unfolded protein response (UPR) activation in *Col3a1*$^{+/G182R}$ mice, corresponding to retention of misfolded proteins in fibroblasts as shown in other fibrillar collagens related diseases [20]. Reverse transcription quantitative PCR (RT-qPCR) experiments using cDNA from thoracic aortas from *Col3a1*$^{+/+}$ (n = 13) and *Col3a1*$^{+/G182R}$ (n = 15) 8-week-old mice

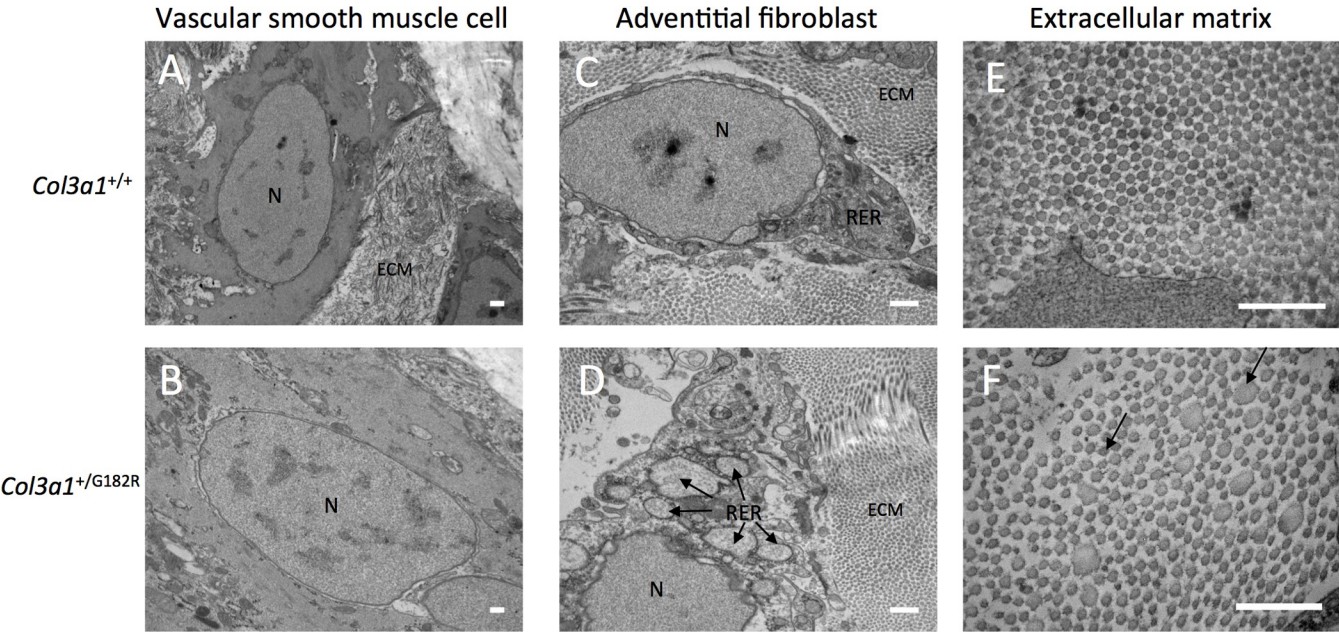

**Fig 3. *Col3a1*$^{+/G182R}$ aortas have abnormal extracellular matrix architecture. A-F** electron microscopy images of the thoracic aorta in 24-week-old *Col3a1*$^{+/+}$ and *Col3a1*$^{+/G182R}$ mice (n = 3 per group). N = nucleus, ECM = extracellular matrix, RER = rough endoplasmic reticulum. Scale bar = 500 nm. **A-B** electron microscopy images of vascular smooth muscle cell in *Col3a1*$^{+/+}$ and *Col3a1*$^{+/G182R}$ mice: normal morphology of vascular smooth muscle cell in *Col3a1*$^{+/G182R}$ mice. **C-D** electron microscopy images of adventitial fibroblasts in *Col3a1*$^{+/+}$ and *Col3a1*$^{+/G182R}$ mice: black arrows indicate dilated RER in adventitial fibroblasts with intense ribosomal activity in *Col3a1*$^{+/G182R}$ mice. **E-F** electron microscopy images of collagen cross fiber diameter in *Col3a1*$^{+/+}$ and *Col3a1*$^{+/G182R}$ mice: important heterogeneity in diameter and decreased density of collagen fibers in *Col3a1*$^{+/G182R}$ mice. Black arrows indicate heterogeneous diameter of collagen fibers.

showed increased expression of Heat shock protein 47 (*HSP47*) and activating transcription factor 6 (*ATF6*) (n = 13 vs. n = 15, p = 0.040 and p = 0.010 respectively, S5 Fig) encoding a specific chaperone of collagens in ER for the former and a ER transmembrane transcription factor sensitive to misfolded proteins for the latter. Besides, increased expression of autophagy protein 5 (*ATG5*) and autophagy protein 7 (*ATG7*) (n = 13 vs. n = 15, p = 0.020 and p = 0.020 respectively, S5 Fig) supports the activation of degradation by autophagy of the misfolded collagen.

## Collagen type III expression

To assess the mRNA expression of both *Col3a1* alleles (WT and mutant) thoracic aortas from *Col3a1*$^{+/+}$ (n = 21) and *Col3a1*$^{+/G182R}$ (n = 18) 24-week-old surviving mice were analyzed using digital droplet PCR. Using this technique, we did not observe any difference in total *Col3a1* mRNA levels between *Col3a1*$^{+/G182R}$ and *Col3a1*$^{+/+}$ mice. WT and mutant *Col3a1* mRNA levels in *Col3a1*$^{+/G182R}$ mice did not differ demonstrating a similar transcriptional expression of both alleles (Fig 4A).

Western blot showed rather the reverse with a trend for increase amount of procollagen type III in arteries of *Col3a1*$^{+/G182R}$ 8-week-old mice (n = 6) compared to their wild-type littermates (n = 6, p = 0.024, Fig 4B). We also assessed the relative amounts of collagen III and collagen I by immunofluorescence on thoracic aorta slices from *Col3a1*$^{+/+}$ (n = 10) and *Col3a1*$^{+/}$

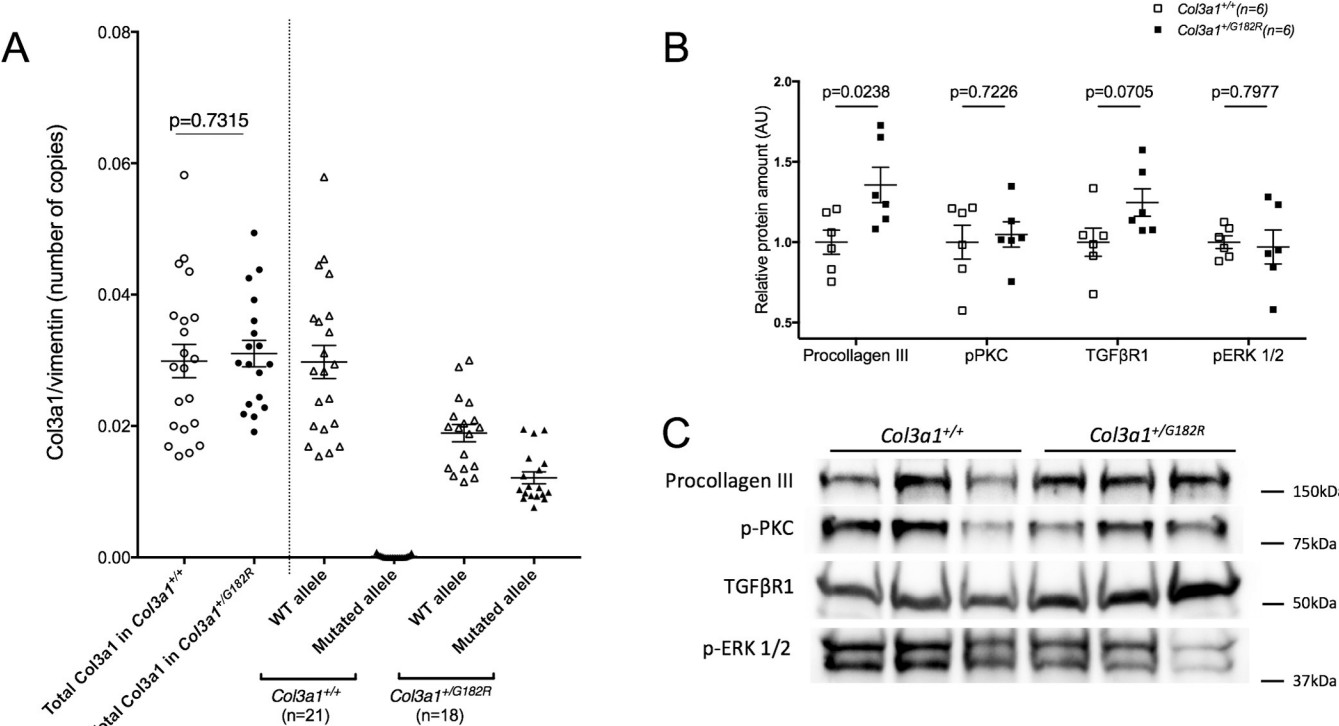

**Fig 4. Expression of *Col3a1* gene and collagen III in aortas of *Col3a1*$^{+/G182R}$ mice. A** Relative allele specific mRNA expression of type III collagen in thoracic aorta of *Col3a1*$^{+/+}$ (n = 21) and *Col3a1*$^{+/G182R}$ (n = 18) 24-week-old mice of both sexes determined by ddPCR. The total expression of the *Col3a1* gene was evaluated (a: unpaired student t-test). The expression of the mutated allele, harboring the Glycine substitution, was not significantly different than the expression of the WT allele in *Col3a1*$^{+/G182R}$ mice, (b: comparative test between theoretical and observed proportions). Furthermore the absence of expression of the mutated allele in *Col3a1*$^{+/+}$ mice confirmed the specificity of the probe of the mutated allele. **B** Quantification of procollagen III, p-ERK, p-PKC, and TGFβ1R levels with western blot analysis, using total protein normalization in *Col3a1*$^{+/G182R}$ (n = 6) and *Col3a1*$^{+/+}$ (n = 6) mice. **C** Corresponding representative Western blot analysis of proteins of TGF-β/Smad and MAPK/ERK signaling pathways: procollagen III, p-ERK, p-PKC and TGFβ1R to compare their expression in *Col3a1*$^{+/+}$ and *Col3a1*$^{+/G182R}$ mice.

*G182R* (n = 10) 7 to 15–week-old mice. Aortic adventitia of *Col3a1*$^{+/G182R}$ showed an apparent drastically reduced expression of mature collagen III (S6A Fig) and consequently a lower expression ratio of collagen III over collagen I (n = 10 vs. n = 10, p$<10^{-4}$, S6B Fig). We deduced that this discrepancy was likely due to a defect in the recognition of the mature mis-folded collagen III rather than a quantitative defect that was not observed at the mRNA and protein levels when using denaturing conditions such as those used in western blotting (SDS-PAGE). Moreover, the two antibodies do not recognize the same antigen, which may explain the discordant results obtained in western blot and immunofluorescence.

## Transcriptome analysis with a focus on MAPK, PLC/IP3/PKC/ERK and TGF-β signaling pathways

To assess the impact of *Col3a1*$^{+/G182R}$ genotype on mRNA expression we performed transcrip-tomic analysis by RNA sequencing on thoracic aortas of *Col3a1*$^{+/+}$ (n = 4) and *Col3a1*$^{+/G182R}$ (n = 3) of 7-week-old male mice. Among 17,475 genes, 210 were differentially expressed genes (DEG) with an adjusted p-value below 0.05 (140 downregulated and 70 upregulated, Fig 5A). The complete gene dataset was analyzed using Gene Set Enrichment Analysis (GSEA, http://software.broadinstitute.org/gsea/index.jsp) to compare transcriptomic data with the "Hall-marks" database, including 50 low-redundancy gene sets of genes involved in important bio-logical processes. We found 13 gene sets enriched in upregulated genes and 1 gene set enriched in downregulated genes in *Col3a1*$^{+/G182R}$ mice (FDR<0.05, Fig 5C). The identified gene sets were essentially related to inflammation ("TNFα signaling via NFκB" and "Inflam-matory response"), and cell stress response ("UV response down", "Hypoxia").

We further focused our analyses on the MAPK pathway notably the PLC/IP3/PKC/ERK signaling pathway activation as previously identified in another vEDS mouse model [19]. First, the comparison was made with the "Canonical pathways" database, which includes 2,887 gene sets from 9 online pathway databases including Kyoto Encyclopedia of Genes and Genomes database (KEGG) https://www.genome.jp/kegg/pathway.html). Normalized enrichment score (NES) of the MAPK signaling pathway was not significantly increased, (NES = 1.63, FWER p value = 0.999). Second, we focused on the 223 genes found in our transcriptomic data among the 267 genes included in the MAPK signaling pathway from KEGG. Analysis found 5 DEGs, among which: *Dusp10*, *Hspa1a*, *Hspa1b*, and *Rps6Ka5* were upregulated, and *Map2k3* was downregulated (Fig 5B and S3 Table).

Finally, we compared the expression of the genes reported by Bowen et al. [19] as belonging to the PLC/IP3/PKC/ERK signaling pathways. We retrieved 36 genes in our dataset among whose: *Hspa1a* and *Hspa1b* were significantly upregulated in *Col3a1*$^{+/G182R}$ mice as previously described [19] and *Ldlr* and *Wwp2* were downregulated, whereas they were reported as upre-gulated in the *Col3a1*$^{G938D/+}$ mice (S4 Table). By western blot on thoracic aortas, we also assessed the expression of p-ERK, p-PKC, and TGFβR1 of the canonical TGFβ pathway. No relative protein amount difference was found except for TGFβR1 which was slightly higher in *Col3a1*$^{+/G182R}$ mice (n = 6 vs. n = 6, p = 0.071, Fig 4B). We further quantified the relative amounts of TGFβ1, TGFβ2 and TGFβ3 using Bio-Plex assay on thoracic aortas of *Col3a1*$^{+/+}$ (n = 6) and *Col3a1*$^{+/G182R}$ (n = 6) of 24-week-old male mice. No significant difference was found for these 3 ligands (S5 Table), this result being consistent with the transcriptomic data (S3 Table).

## Impact of different antihypertensive agents on survival rate

Since arterial blood pressure is a key determinant of the arterial wall stress, we investigated the effects of different types of antihypertensive drugs on the survival rate, systolic BP (SBP) and

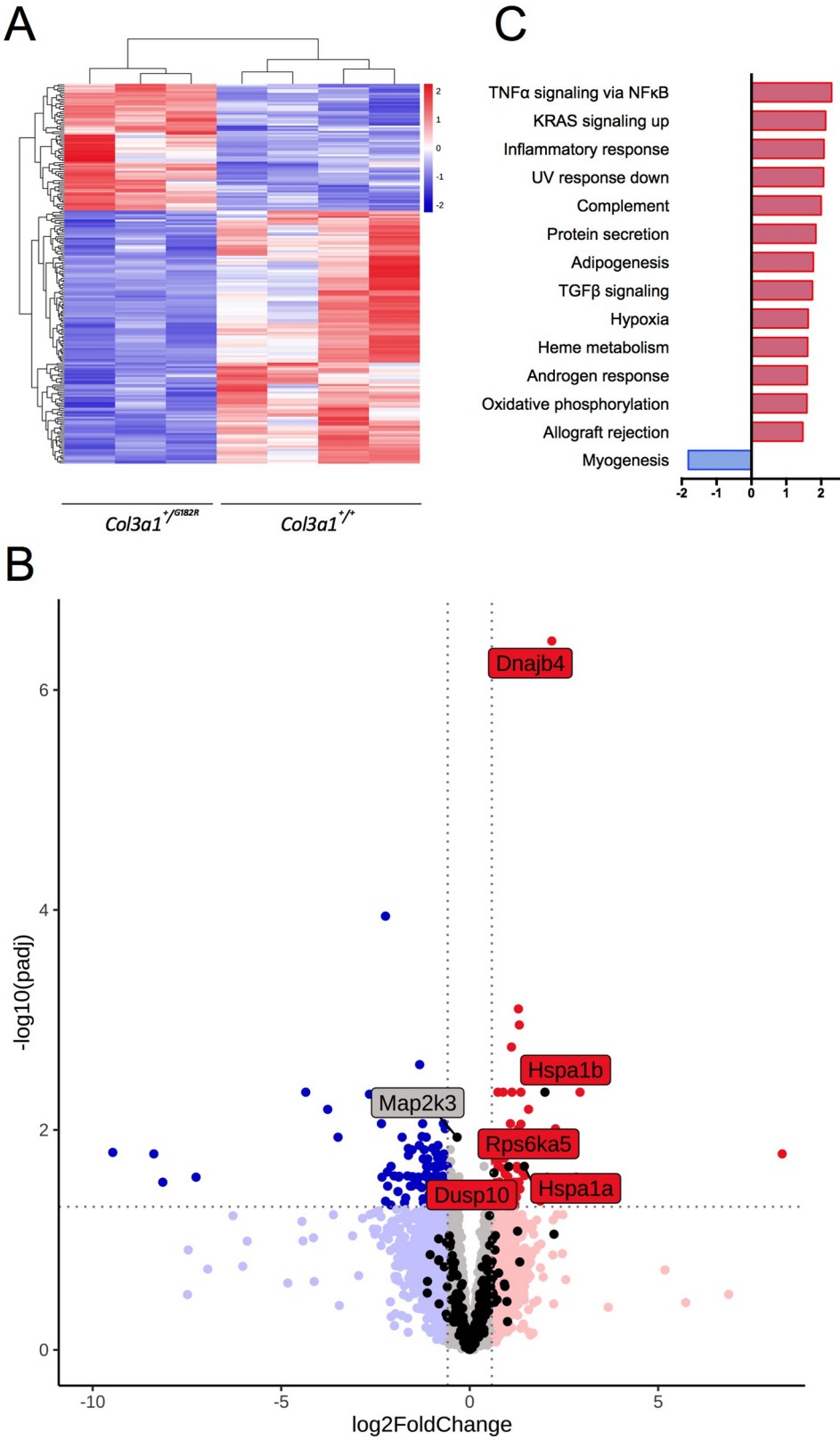

**Fig 5. The p.G182R Col31a variation results in inflammation and cell stress response and normal PLC/IP3/PKC/ERK signaling pathway in TA transcriptome. A** Unsupervised hierarchical clustering using the 210 differentially

expressed genes (DEG, adjusted p<0.05) from RNAseq in *Col3a1*$^{+/G182R}$ (n = 3) and *Col3a1*$^{+/+}$ (n = 4). **B** Volcano plot of the complete gene dataset in descending TA. Grey represents the genes that were not modified; light blue, the genes with insignificant expression decreased more than 1.5-fold; dark blue, the DEGs with expression decreased more than 1.5-fold; light red, the genes with insignificant expression increased more than 1.5-fold, dark red, the DEGs with expression increased more than 1.5-fold; black, the 223 genes in the dataset and belonging to the MAPK signaling pathway from the KEGG database. The 5 DEG of the MAPK signaling pathway were labelled indicated their name. The sixth indicated gene Dnajb4 does not belong to this pathway. **C** Normalized enrichment score (NES) of the 14 enriched gene sets from the "hallmarks" database in GSEA. Positive and negative values indicate the upregulated and downregulated genes respectively in the *Col3a1*$^{+/G182R}$ mice in comparison to *Col3a1*$^{+/+}$ mice (controls).

heart rate (HR) in *Col3a1*$^{+/G182R}$ male mice up to 24-week of age (for details at each time points on SBP and HR, see S6 an S7 Tables).

## Betablockers

As expected, propranolol significantly lowered HR (486.1 ± 17.8 vs. 633.9 ± 28.6 bpm, p = $3.31 \times 10^{-5}$) but did not decrease SBP (a significant increase was even observed at 24 weeks, 115.7 ± 3.6 vs. 106.6 ± 2.8 mmHg, p = 0.002). No benefit on the 24-week mortality rate was observed compared to water (40.0% vs. 46.7%, p = 0.686; Fig 6A and 6B). As the betablocker celiprolol showed benefits in vEDS patients [10,11], we tested whether this drug could have any effect on aortic rupture in *Col3a1*$^{+/G182R}$ mice especially because of its specific pharmaco-dynamic profile adding a β(1)-adrenoceptor antagonist to a β(2)-adrenoceptor agonist action [21]. At a dose of 250 mg/kg/day, celiprolol changed neither the SBP (116.5 ± 4.8 vs. 116.6 ± 4.8 mmHg, p = 0.378, throughout the 24-week follow-up period) nor the HR (624.5 ± 16.9 vs. 628.8 ± 22.9 bpm, p = 0.634). No benefit on the 24-week mortality rate was observed compared to water (50.0% vs. 36.8%, p = 0.541; Fig 6C and 6D). We checked that plasma concentration of celiprolol at 24 weeks (115.0 ± 15.0 mg/ml) (n = 80 mice, S7 Fig) was above the level usually recorded in patients taking 200 to 400 mg of celiprolol per day.

## Calcium channel blockers and hydralazine

Amlodipine is a long-acting calcium channel blocker inhibiting calcium ions flux into vSMC, thus causing vasodilation, reduction of peripheral vascular resistance and BP decrease [22]. Giving an average daily dose of 7.5 mg/kg resulted in a SBP decrease (107.5 ± 4.9 vs. 125.8 ± 4.6 mmHg), accompanied by an increase in HR (733.3 ± 16.9 vs. 683.6 ± 18.6 bp/min), only significant at 16 weeks (Fig 7B and S6 Table). No benefit on mortality rate was observed at 24 weeks compared to water (75.0% vs. 60.9%, p = 0.126, Fig 7A), with even a trend toward a worsening mortality at the age of 15 weeks (p = 0.068). We hypothesized that the stimulation of the sympathetic nervous system that follows the administration of a peripheral vasodilating agent might have blunted its possible benefit [23]. We thus tested the addition of propranolol to amlodipine aiming at blocking the sympathetic stimulation. A significant effect of the association was observed on both SBP and HR compared to amlodipine alone (SBP: 98.5 ± 3.6 vs. 107.5 ± 4.9 mmHg, p = 0.027; HR: 604.9 ± 20.2 vs. 733.3 ± 16.9 bpm, p = $2.13 \times 10^{-6}$, throughout the 24-week follow-up period, Fig 7B). Adding propranolol allowed to reverse the increased mortality at 15 weeks with amlodipine alone, but did not induce a significant benefit on the 24-week mortality rate compared to amlodipine alone (70.0% vs. 75.0%, p = 0.340, Fig 7A). Similar patterns were observed in female mice (S8 Fig and S7 Table).

We also tested hydralazine a direct-acting vasodilator, decreasing peripheral vascular resistance and blood pressure, known to elicit a reflex sympathetic stimulation of the heart [24]. Hydralazine was also recently proposed by Bowen et al. [19] as improving survival rate of a similar mouse vEDS model, possibly by an inhibiting effect on ERK pathway [19]. In our

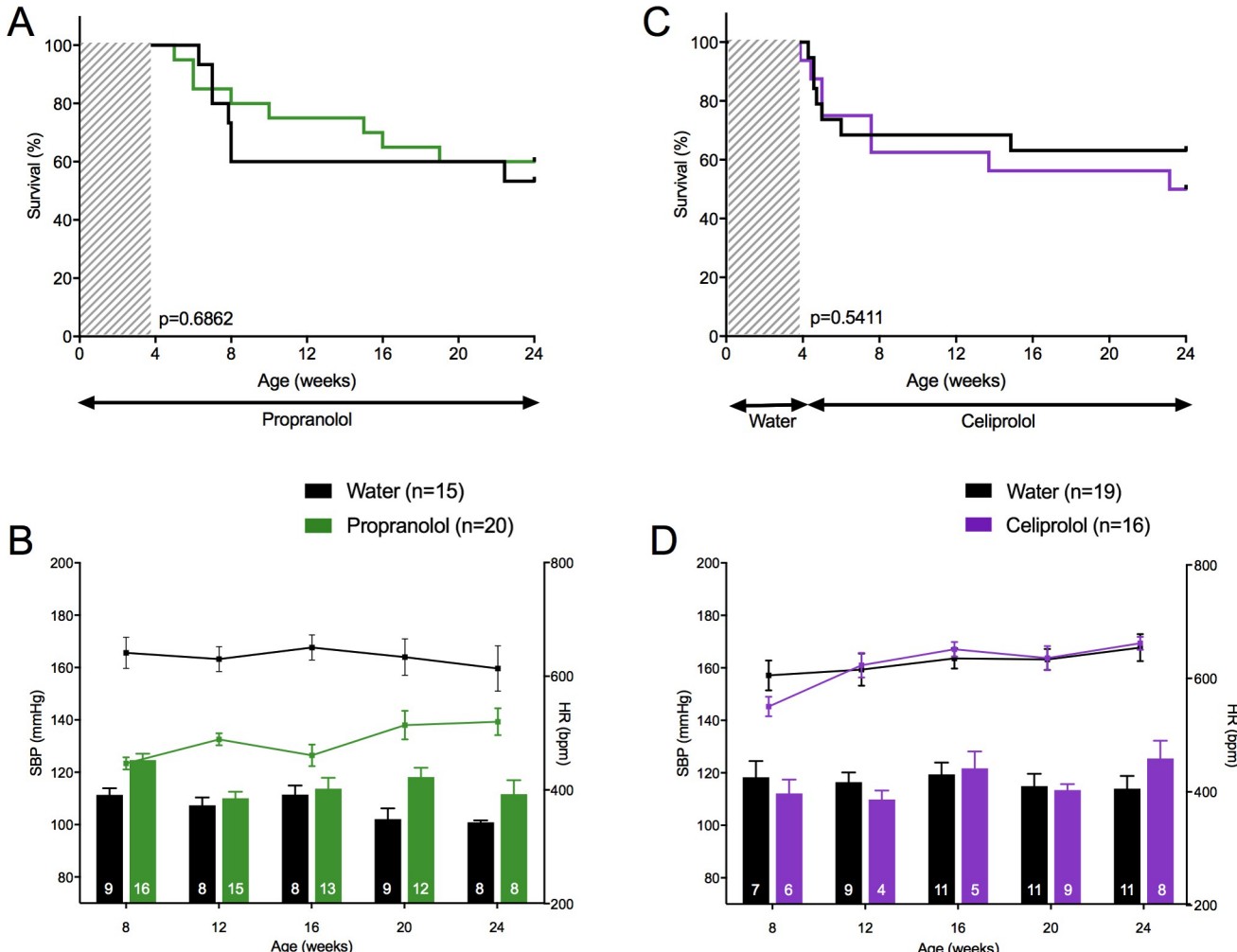

**Fig 6. Therapeutic challenges: hemodynamic parameters and survival curves of β-blockers.** The upper panels represent the survival curves on either active treatment or water. The lower panels represent SBP (lower bars: left scale) and HR (upper lines: right scale), measured between 8 and 24 weeks. Data are expressed as mean ±SEM. Numbers within the bars indicate the number of living mice studied at each time of measurement. **A-B** Comparison between propranolol and water. **A** Kaplan-Meier Survival curve for comparing *Col3a1*$^{+/G182R}$ treated with propranolol (n = 20) to *Col3a1*$^{+/G182R}$ treated with water (n = 15). Insignificant difference is calculated using Log-Rank (Mantel-Cox) analysis (p = 0.6862). **B** Despite a significant decrease in HR (student t-test, p<0.05 at each time except 24 weeks), there was no decrease in SBP on propranolol that was even higher than in the untreated group (student t-test, p<0.05 at 8 and 20 weeks). **C-D** Comparison between celiprolol and water. **C** Kaplan-Meier Survival curve for comparing *Col3a1*$^{+/G182R}$ treated with celiprolol (n = 16) to *Col3a1*$^{+/G182R}$ treated with water (n = 19). Insignificant difference is calculated using Log-Rank (Mantel-Cox) analysis (p = 0.5411). **D** SBP and HR were not changed by celiprolol (student t-test, p>0.05 throughout the follow-up period).

model, hydralazine significantly increased HR (701.8 ± 12.4 vs. 639.4 ± 15.1 bpm, p = 1.69×10$^{-8}$), but did not influence the SBP level (115.1 ± 2.8 vs. 119.6 ± 3.3 mmHg, p = 0.236, Fig 8B). A non-significant decrease of the 24–week mortality rate was observed (29.9% vs. 50.0%, p = 0.136, Fig 8A). In order to block the sympathetic stimulation we chose to combine celiprolol to hydralazine. No significant effect of the association was observed on both SBP or HR compared to hydralazine alone (p = 0.359 and p = 0.061, respectively, Fig 8B). No benefit of the association on the 24-week mortality rate was observed compared to hydralazine alone, the mortality rate superposing with the one observed in non-treated mice (46.7% vs. 50.0%, p = 0.848, Fig 8A).

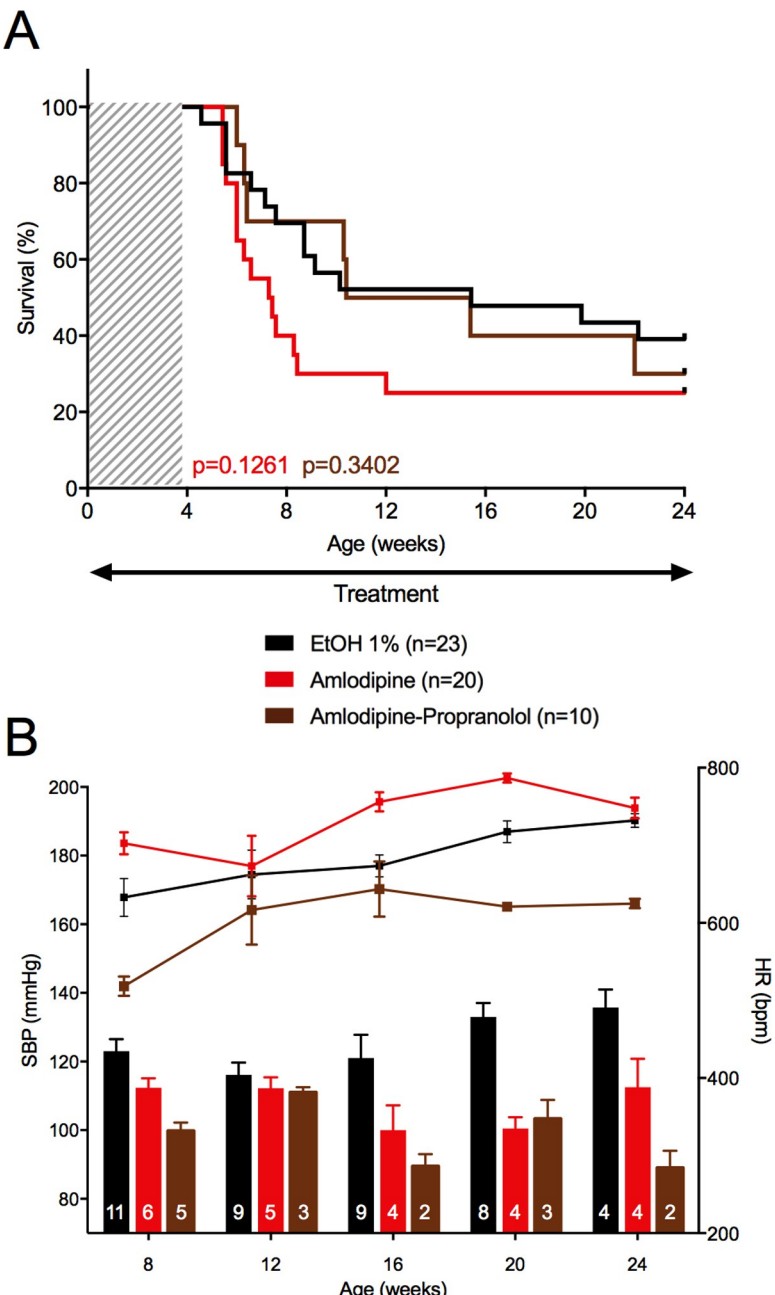

**Fig 7. Consequences of treatment by amlodipine and the association amlodipine-propranolol.** The upper panel represents the survival curves on either active treatment or water. The lower panels represent SBP (lower bars: left scale) and HR (upper lines: right scale), measured between 8 and 24 weeks. Data are expressed as mean ±SEM. Numbers within the bars indicate the number of living mice studied at each time of measurement. **A** Survival comparison between Amlodipine and ethanol 1%, and between the association amlodipine-propranolol and amlodipine (monotherapy). Kaplan-Meier Survival curve (red curve) for comparing $Col3a1^{+/G182R}$ treated with amlodipine (n = 20) to $Col3a1^{+/G182R}$ treated with ethanol 1% (n = 23). Insignificant difference is calculated using Log-Rank (Mantel-Cox) analysis (p = 0.1261). Kaplan-Meier Survival curve (brown curve) for comparing $Col3a1^{+/G182R}$ treated with amlodipine-propranolol (n = 10) to $Col3a1^{+/G182R}$ treated with amlodipine (n = 20). Insignificant difference is calculated using Log-Rank (Mantel-Cox) analysis (p = 0.3402). **B** SBP and HR comparison between amlodipine and ethanol 1%, and between the association amlodipine-propranolol and amlodipine (monotherapy). SBP was slightly lower in the treated group (student t-test, p>0.05 throughout the follow-up period except at 20 weeks) and HR was slightly increased (student t-test, p<0.05 at 16 and 20 weeks) when comparing amlodipine to ethanol 1% (red bars and curve). SBP remained unchanged throughout the follow-up period (Student-t test, except at 8 weeks, p = 0.0086) and HR decreased significantly (student t-test, p<0.05 throughout the follow-up period, except at 12 weeks) when comparing amlodipine-propranolol to amlodipine (monotherapy, brown bars and curve).

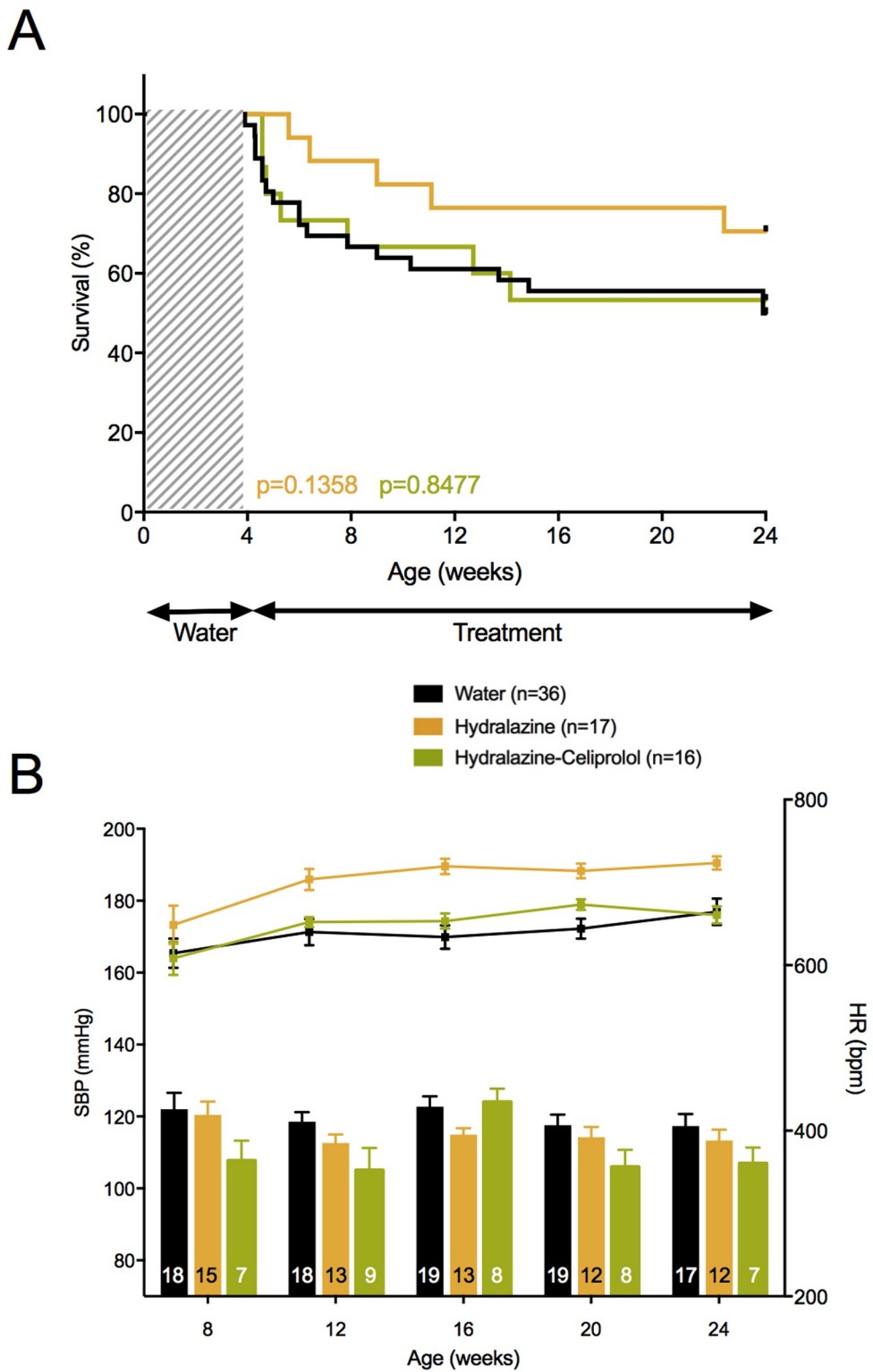

**Fig 8. Consequences of treatment by hydralazine and the association hydralazine-celiprolol.** The upper panels represent the survival curves on either active treatment or water. The lower panels represent SBP (lower bars: left scale) and HR (upper lines: right scale), measured between 8 and 24 weeks. Data are expressed as mean ±SEM. Numbers within the bars indicate the number of living mice studied at each time of measurement. **A** Survival comparison between hydralazine and water, and between the association hydralazine-celiprolol and water. Kaplan-Meier Survival curve (orange curve) for comparing $Col3a1^{+/G182R}$ treated with hydralazine (n = 17) to $Col3a1^{+/G182R}$ treated with

water (n = 36). Insignificant difference is calculated using Log-Rank (Mantel-Cox) analysis (p = 0.1358). Kaplan-Meier Survival curve (green curve) for comparing $Col3a1^{+/G182R}$ treated with hydralazine-celiprolol (n = 16) to $Col3a1^{+/G182R}$ treated with water (n = 36). Insignificant difference is calculated using Log-Rank (Mantel-Cox) analysis (p = 0.8477). **B** SBP and HR comparison between hydralazine and water, and between the association hydralazine-celiprolol and water. No significant decrease of SBP was observed and significant decrease of HR (student t-test, p<0.05 from 12 weeks) was observed between hydralazine treated and water groups. SBP and HR were not changed by hydralazine-celiprolol (student t-test, p>0.05 throughout the follow-up period).

## Angiotensin II type 1 receptor antagonist

We then tested losartan, an Ang II type 1 receptor blockade whose benefits have been well established in cardiovascular diseases and discussed in genetic aortic diseases [25]. Started during breeding, losartan drastically lowered SBP (–24.9 mmHg on average; p = $2.64 \times 10^{-7}$, S6 Table) and markedly decreased the 24–week mortality rate compared to non–treated mice (17.6% vs. 62.5%, p = 0.021, Fig 9A). To test whether this beneficial effect was reversible, losartan was stopped at 24 weeks with another 24-week follow-up. Mortality due to spontaneous aortic ruptures reappeared within the next 10 weeks. At 48 weeks of age, the mortality rate was similar between previously losartan-treated mice and non-treated mice (58.8% vs. 62.5%, p = 0.415, Fig 9A).

Because of the activation of the renin angiotensin system during the neonatal period [26], we tested whether this beneficial effect would remain when starting the drug in young adult mice. Started at weaning (4 weeks of age), losartan lowered significantly SBP (–17.7 mmHg on average; p = 0.004, Fig 9B), did not change the HR and markedly decreased the 24–week mortality rate compared to non–treated mice (7.7% vs. 62.5%, p = 0.003; Fig 9C). Blood pressure effect of losartan was confirmed by telemetry measurements (S9 Fig).

To assess if the activation of the renin angiotensin system could explain the efficiency of losartan in $Col3a1^{+/G182R}$ mice, we found 6 genes belonging to the renin angiotensin system in our transcriptomic dataset and none of them was significantly upregulated (S8 Table).

The impact of the renin angiotensin system in this model was also tested by the administration of Ang II at doses with minimal effect on SBP as we already reported in $Col3a1$ htz knock-out mice [14]. Ang II was administered subcutaneously at a dose of 0.5μg/kg/min during 14 days in 24-weeks of age surviving male mice. The deleterious effect of Ang II was major and fast with a 100% mortality rate within 10 days in $Col3a1^{+/G182R}$ mice vs. 22.2% in $Col3a1^{+/+}$ mice (p<$10^{-4}$, Fig 9D).

## Discussion

Patients suffering from vEDS have a life expectancy mainly reduced by the occurrence of unpredictable arterial rupture in young adulthood. Despite randomized open clinical trial [10] and cohorts surveys [11,27] data, there is no consensus on a continuous medical therapy that could prevent that risk [2,28]. In such a rare disease, the development of pre-clinical models is particularly useful to test different therapies and better understand the consequences of the disease at the cellular and tissue levels.

We report herein a new knock-in mouse model of vEDS developed on a C57BL/6J genetic background which recapitulates most of the vEDS features in humans. Heterozygous $Col3a1^{+/G182R}$ mice suffered from spontaneous mortality in early adulthood mainly by aortic rupture and among homozygous $Col3a1^{G182R/G182R}$ mice which were born, perinatal mortality was dramatic. This is in line with what is observed in humans, patients being heterozygous at the $COL3A1$ gene, and very rare more syndromic and severe cases with biallelic variants [29]. We also confirmed a strong mortality associated with pregnancy and lactation [18,19], and most importantly, a strong sexual difference, underlined by a much higher mortality in male mice

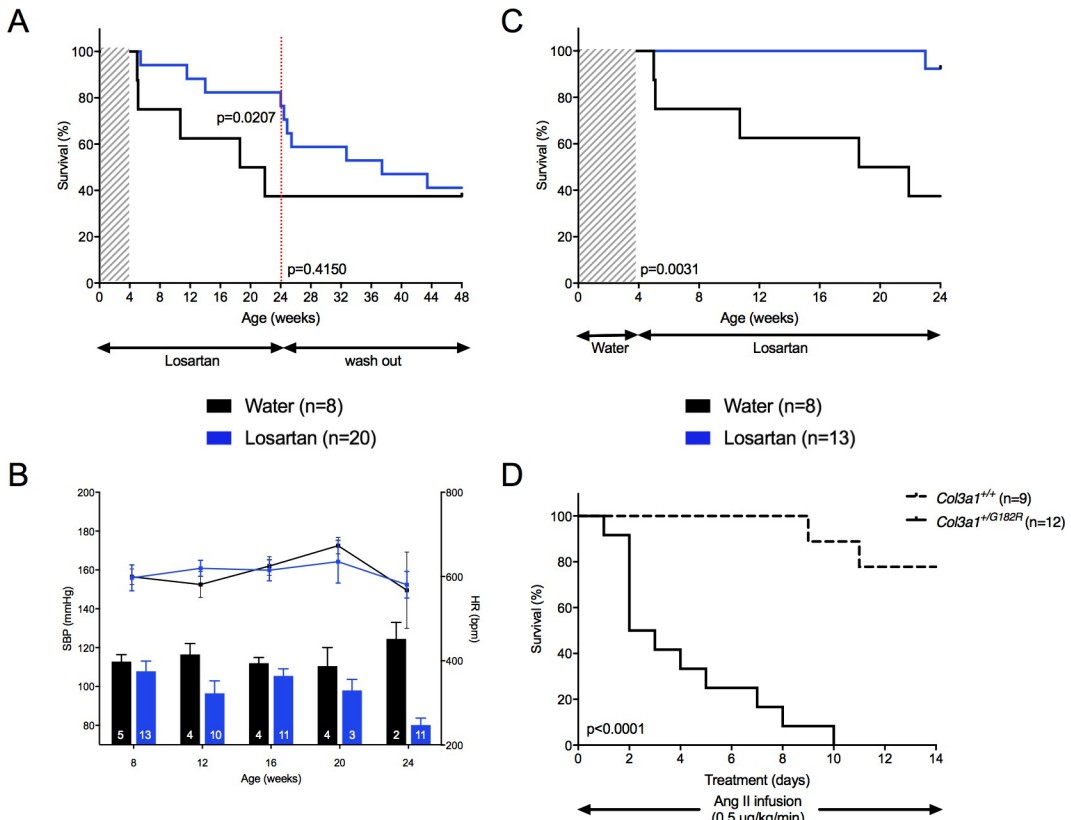

**Fig 9. Consequences of on- and off-treatment by losartan.** The upper panels represent the survival curves on either active treatment or water. The lower panels represent SBP (lower bars: left scale) and HR (upper lines: right scale), measured between 8 and 24 weeks. Data are expressed as mean ±SEM. Numbers within the bars indicate the number of living mice studied at each time of measurement. **A** Comparison between losartan started in pregnant mothers and water. Kaplan-Meier Survival curves for comparing *Col3a1*$^{+/G182R}$ mice (n = 20) with a 24-week period of losartan followed by a 24-week washout period and *Col3a1*$^{+/G182R}$ (n = 8) with no treatment. Significant difference in mortality in treated mice (18% mortality at 24 weeks) compared to the untreated group (63% mortality at 24 weeks) is calculated using Log-Rank (Mantel-Cox) analysis (p = 0.0207) at the end of the 24-week treatment. When the administration of losartan was stopped, an important increased mortality was observed and survival rates at 48 weeks of age were comparable between the 2 groups of mice (41% vs 38%, p = 0.4150). **B-C** Comparison between losartan started at weaning and water. B SBP was significantly changed by losartan (student t-test, p<0.05 at 24 weeks of the follow-up period) and HR was not changed by losartan (student t-test, p>0.05 throughout the follow-up period). **C** Kaplan-Meier Survival curve for comparing *Col3a1*$^{+/G182R}$ treated with Losartan started at age 4 weeks (n = 13) to *Col3a1*$^{+/G182R}$ treated with water (n = 8). Losartan significantly improve the survival using Log-Rank (Mantel-Cox) analysis (p = 0.0031). **D** Kaplan-Meier Survival curve for comparing *Col3a1*$^{+/+}$ (n = 9, doted curve) to *Col3a1*$^{+/G182R}$ (n = solid curve) male mice with Angiotensin II infusion (0.5 µg/kg/min). Significant difference is calculated using Log-Rank (Mantel-Cox) analysis (p<0.0001).

than in out of breeding female mice. Such a sexual difference was also observed by Bowen et al. in their *Col3a1* knock-in model [19] echoing the possible excess mortality of young vEDS male patients, as reflected by the slight female predominance in adult cohorts [11]. Whereas a marked sexual difference exists in human more common cardiovascular diseases leading to the concept that testosterone might be detrimental in their development, the underlying mechanisms behind this hormonal deleterious effects in case of collagen III deficiency warrant further studies.

Arterial investigations of our *Col3a1*$^{+/G182R}$ mice model gave further insights into the pathophysiological mechanisms at play on the aortic wall in vEDS. First, we could confirm by echography that dilatation or aneurysm did not precede aortic rupture, contrary to what can be observed in the Marfan mouse model or the corresponding human pathology [30]. Second,

we observed a lower collagen content in the aortic wall along with a smaller adventitia area and a preserved intima-media thickness. In addition, no collagen III in its native conformation could be detected in aortas of $Col3a1^{+/G182R}$ mice. This observation is coherent with old experiments showing that vessels treated with collagenase are prone to rupture and that wall integrity depends on intact collagen [31].Thinning and collagen content reduction of the adventitia are in line with findings from other vEDS knock-in mouse models [18,19]. In $Col3a1^{+/G182R}$ mice, Goudot et al. showed a reduced aortic stiffening over the cardiac cycle and confirmed the impaired biomechanical aortic properties of this model, particularly when pressure rises and collagen fibers are mechanically requested [32]. Similar results were found in vEDS patients in comparison with controls [9] which strengthens the relevance of our mouse model to recapitulate vEDS features.

Recently the PLC/IP3/PKC/ERK pathway has been proposed as a major contributor to the arterial pathogenesis in $Col3a1^{+/G938D}$ mice [19]. We did not observe such activation both at the mRNA and protein levels in our model. The identified gene sets in our transcriptomic data obtained from aortic tissue were essentially related to a mild increase in inflammation signaling ("TNFα signaling via NFκB" and "Inflammatory response"), and cell stress response ("UV response down", "Hypoxia") and not to the PLC/IP3/PKC/ERK pathway. Similarly, western blots showed neither significant increase in p-ERK nor p-PKC in $Col3a1^{+/G182R}$ mice compared to $Col3a1^{+/+}$ littermates. In addition, the canonical (Smad2/3) TGFβ-dependent signaling cascade does not seem to be responsible for the vascular fragility of our knock-in model with consistent results of both western blot and Bio-plex analyses. Thus, the contribution of the noncanonical (ERK1/2) TGFβ-dependent signaling cascade remains to be elucidated. In that regard, we tested hydralazine whose potential benefit has been partly explained by the inhibition of the IP3-mediated calcium release from the ER and hence PKCβ activation [33]. Only a trend toward increased survival but still not significant, was observed in $Col3a1^{+/G182R}$ mice treated with hydralazine.

One could also explain the absence of clear beneficial effect of hydralazine by the stimulation of the sympathetic nervous system secondary to the peripheral arterial dilation, as reflected by the 10% HR increase in hydralazine-treated mice. Therefore, we tested amlodipine, a long-acting calcium channel blocker causing direct vasodilation and improving arterial stiffness of hypertensive patients [34]. A 7% HR increase was associated with an 18 mmHg decrease in BP at 24 weeks. Despite this BP lowering effect, amlodipine induced a worsening of the mortality rate in $Col3a1^{+/G182R}$ mice. In order to counteract the sympathetic activation, we tested whether the addition of a beta-blocker could prevent this increased mortality. The combination of propranolol to amlodipine reversed the increased mortality observed at 15 weeks in $Col3a1^{+/G182R}$ mice treated with amlodipine alone but did not improve survival compared to non-treated animals. This suggest that the stimulation of the sympathetic nervous system could, only in part, be responsible for the deleterious effect of amlodipine. Interestingly, reduced survival rate with amlodipine was also observed by Dietz's group in Marfan [35] and vEDS mice [19]. Their explanation was that amlodipine enhanced activation of both canonical (Smad) and noncanonical (ERK1/2) TGFβ-dependent signaling cascades, contrary to hydralazine acting as an ERK inhibitor which had a beneficial effect in their hands. They proposed that one main mechanism leading to vascular rupture in vEDS is an excessive signaling through the PLC/IP3/PKC/ERK axis. Using the same drugs daily doses, we could not replicate these findings even though our opposite results of amlodipine and hydralazine are in part consistent with their hypothesis.

Because of their BP lowering effects associated with a reduction of sympathetic activity, beta-blockers have been largely used in the prevention and treatment of cardiovascular diseases. They have also been proposed in inherited aortic aneurysms such as Marfan syndrome,

but their benefits on aortic root dilation and the incidence of cardiovascular surgery are disputed [36]. In vEDS, only celiprolol has been rigorously tested [37] showing a significant benefit on cardiovascular mortality and morbidity on a limited number of molecularly-proven patients [10]. Its good tolerance and beneficial effects were also shown in a French cohort with long-term survey (9 years) [11], and a 2-year follow-up of 40 Swedish patients [27]. The significance of these results being disputed, we tested both propranolol, the oldest non-selective β-blocker generally used in experimental studies, and celiprolol (at therapeutic plasma concentration). Both β-blockers had no beneficial effect on $Col3a1^{+/G182R}$ mice survival. In the study reported by Bowen et al. [19], celiprolol even accelerated the rate of death from aortic dissection in their two $Col3a1$ knock-in mouse models, whereas propranolol had neutral effect [19]. Overall, these results should be interpreted cautiously since therapeutic results observed in mouse cardiovascular models are not always reproduced in humans, as demonstrated in Marfan syndrome [38].

Our major therapeutic findings were obtained when targeting the renin angiotensin system. Losartan improved markedly the survival of $Col3a1^{+/G182R}$ and was associated with a strong SBP decrease (–25 mmHg). That major BP effect could in part explain its strong beneficial effects. It is noteworthy that stopping losartan induced a return to increased mortality by aortic ruptures close to the non-treated group and that beginning losartan after weaning had the same strong beneficial effect on survival. Conversely, the administration of Ang II at doses with minimal effect on SBP (0.5 μg/Kg/min) induced aortic rupture and a 100% mortality rate within the next 10 days after administration. The multiple actions of Ang II as well as the role of Ang II type 1 receptor (AT1R) on vascular tone and cellular growth are consistent with these results. Indeed, structural vascular remodeling induced by Ang II involves multiple signaling pathways including canonical (Gq/PLC/PKC) and noncanonical pathways such as transactivation of the EGFR receptor, ROS production, Jak2 and c-src activations [39]. Finally, AT1R modulate sympathetic vasomotor function [40] and we showed that the stimulation of the sympathetic nervous system could play a deleterious role in our model. Thus, despite that no significant benefit of losartan (at the same daily dose) was obtained by Dietz's group on survival rate of their $Col3a1^{+/G938D}$ mouse model [19], we firmly believed that the use of Ang II receptor antagonist should be tested in vEDS patients in a randomized controlled trial (https://clinicaltrials.gov/ct2/show/NCT02597361?cond=Vascular+Ehlers-Danlos+Syndrome&cntry=FR&draw=2&rank=4).

To conclude, the full clinical, biochemical and pharmacological investigation performed in our new vEDS knock-in mouse model showed a sexual difference on survival rate, the presence of thin non-inflammatory arteries, a cellular phenotype showing enlarged endoplasmic reticulum and unfolded protein response, and no clear activation of the ERK pathway. The spontaneous occurrence of aortic rupture led to test several anti-hypertensive agents based on the reasoning that any stabilization or decrease in blood pressure would reduce arterial wall stress and improve survival rate. Our main results are that betablockers seem to be ineffective, dihydropyridines responsible for possible worsening, whereas treatment based on Ang II receptor blockade shows a great benefit and administration of Ang II induces a rapid mortality caused by aortic rupture. Thus, the renin angiotensin system seems to play a key pathophysiological role in this model, suggesting strongly to test the use of Ang II antagonists in vEDS patients.

## Materials and methods

### Ethics statement

The care and treatment of animals followed the N° 2010/63/UE European legislation and national authority (Ministère de l'agriculture, France) guidelines for the detention, use, and

ethical treatment of laboratory animals. All the experiments were approved by the local ethics committee (MESR 20714 approval number) and experiments were conducted by authorized personnel.

## Generation of the targeting vector

A Bacterial Artificial Chromosome (BAC) was used to generate the homologous recombination vector using the recombineering technology described by the group of N. Copeland [41]. This BAC (bMQ223m12) was identified *in silico* within the library constructed from the genomic DNA of the AB2.2 ES cell line [42], using the Ensembl genome browser (www.ensembl.org). The chosen BAC contained the whole *Col3a1* gene, 14 kb sequence upstream and 103 kb sequence downstream of the *Col3a1* gene. It presented the expected restriction profile after electroporation in EL250 bacteria [41]. We modified the BAC as briefly described below to create the targeting vector. The sequence of the primers used to amplify by PCR all the homology arms is given in S1 Table.

First, exon 6 of *Col3a1* gene was replaced by an Ampicillin resistance ($Amp^r$) cassette by homologous recombination in bacteria. The $Amp^r$ cassette was then replaced by a mutated exon 6 sequence, containing the c.547G>A mutation introduced by site-directed mutagenesis (QuickChange Lightning kit, Agilent Technologies), and a *pgk-neo$^r$* selection cassette flanked by FRT sites, located upstream of the mutated exon 6. After these two homologous recombination events, the BAC contained respectively 28 kbp and 136 kbp of sequence homology arms upstream and downstream of exon 6. Therefore, we decided to reduce the size of the BAC in order to facilitate the identification of recombinant ES cell clones by Southern blot, using the gap-repair strategy [41], in the pBluescript II SK plasmid (Agilent Technologies). The final targeting vector thus contained 4.8 kbp upstream and 7.2 kbp downstream of *Col3a1* exon 6.

## Generation of ES cell clones

Twenty µg of the targeting vector were linearized by enzymatic restriction with *Acc651* and electroporated into CK35 ES cells by the "Service des Animaux Transgeniques" (SEAT, Villejuif, France). After a 2-week selection with G418, individual clones were picked. 180 ES cell clones were screened to confirm correct homologous recombination by Southern blot. For that purpose, the genomic DNA of ES cells was extracted and submitted to enzymatic restriction with *BamH1*. We used a probe located upstream of the 5' homology arm, with predicted bands of 12.8 kb and 8.9 kb for the WT and targeted *Col3a1* allele respectively. Five of the 180 ES clones were positive. Additionally, the presence of the G to A mutation in exon 6 was verified by sequencing the genomic DNA of the positive ES clones.

## Generation of *Col3a1$^{+/G182R}$* animals

Two positive ES cell clones were microinjected into C57/BL6J blastocysts and implanted into pseudo-pregnant females using standard techniques. Chimeric founders were crossed with C57/BL6J mice. The resulting animals were genotyped using the following PCR primers: Col3a1KI-fw 5'-TCATCTGAAGTAAAGTTTTCATGC-3', Col3a1KI-rv 5'-TTTCACCGA AATTGAGTGGTT-3' and pgkR1 5'-GGGGAGGAGTAGAAGGTGGCGCGAA-3', to generate PCR products of 468 and 323 bp for WT and targeted mice, respectively. The htz knock-in mice were bred with *hACTB-FLPe* transgenic mice [43] to delete the *FRT-pgk-neo$^r$-FRT* cassette. The animals bearing the *Col3a1$^{+/G182R}$* allele deleted from the neo cassette were identified by PCR with the Col3a1KI-s and Col3a1KI-as primers (480 bp). Resulting *Col3a1$^{+/G182R}$* mice were crossed with C57/BL6J mice to amplify the line and eliminate the *hACTB-FLPe* transgene. The absence of the *hACTB-FLPe* transgene was verified by PCR with the following

primers: oIMR1348 5'-CACTGATATTGTAAGTAGTTTGC-3' and oIMR1349 5'-CTAGTGC GAAGTAGTGATCAGG-3'. Mice carrying the *col3a1*$^{+/G182R}$ allele but not the *hACTB-FLPe* transgene were selected for further breeding.

## Animal studies

**Survival rate.**   The welfare and the mortality were checked daily. For dead mice, the cause of death was determined, and particularly hemothorax and hemoperitoneum were noticed.

**Angiotensin II infusion.**   Ang II (dissolved at the dose of 0.5μg/kg/min in saline solution) was used to fill osmotic mini-pump (model ALZET 1002) as previously described [44]. *Col3a1*$^{+/G182R}$ and *Col3a1*$^{+/+}$ male mice had the same age (31-week-old mice) and the concentration of Ang II was adjusted in osmotic mini-pump because weights were significantly different (from 26 g to 34 g, p<0.05 between groups for weights comparison within each single series). Filled mini-pumps were, before implantation, equilibrated 24 hours in saline solution, at 37˚C. They were then implanted subcutaneously in the back of the mouse previously anesthetized with isoflurane (2%). Surviving mice were all sacrificed at day 14 of Ang II infusion.

**Systolic blood pressure and heart rate measurement.**   Systolic Blood Pressure (SBP) and Heart Rate (HR) were measured on conscious *Col3a1*$^{+/G182R}$ and *Col3a1*$^{+/+}$ mice using a tail cuff system (BP-2000 Visitech Systems) as described previously [45].

Measurements were always performed at the same hour of day in an experiment. For each animal, the system automatically performed 4 first measurements that were not recorded; then, 10 consecutive measurements of SBP and HR were recorded for each mouse (leading to 10 or less SBP and HR values for each mouse). For our analysis, we only kept SBP and HR measurements for which we had at least 4 (out of 10) values. To avoid procedure-induced anxiety, and for each series of experiments, mice were initially accustomed to the tail cuff system during minimum 5 consecutive days. Then, for each series of experiments, SBP and HR were measured at weeks 8, 12, 16, 20 and 24.

SBP, HR and activity were also measured using Radiotelemetry monitoring. The analysis was performed on *Col3a1*$^{+/+}$ and *Col3a1*$^{+/G182R}$ male mice (11-week-old to 40-week-old). Mice were implanted with BP and HR measuring telemetric probes (DSI).

Briefly, the monitoring system consists of a transmitter (PA-C10, Data Sciences International), receiver panel, consolidation matrix and personal computer with accompanying software. Mice were anesthetized with Ketamine/xylasine. A small incision was made in the middle of neck for insertion of the telemetry transmitter and a flexible catheter of the transmitter was surgically placed in the isolate left carotid artery and advanced down to the aortic arch. The body of the transmitter was placed subcutaneously in the right ventral flank of the animal. Mice were allowed recovering for 2 weeks in individual cages. Baseline values for SBP, HR and activity parameters were recorded for 4 consecutive days, every 30 minutes for 30 seconds. After baseline recording, animals were treated with losartan (135 mg/kg/day, dissolved in drinking water). Under challenge, SBP, HR and activity were continuously monitored to visualize effects of the drug for 4 consecutive days.

**Tissue collection and histology.**   Depending on the type of procedure, the mice were sacrificed by cervical dislocation or lethal dose of ketamine/xylazine injected intraperitoneally. If plasma was required, blood was collected by intracardial puncture. To collect the TA, a midsternal thoracotomy was performed and TA was carefully exposed and excised. Distal part (± 5 mm) was fixed in 4% paraformaldehyde for 24h then embedded in paraffin, proximal part (± 10 mm) was degreased, frozen in liquid nitrogen and stored at −80˚C. A piece of tail was sampled too from each animal to perform a new genotyping as a control. All histological analyses were performed from cross-sections of 5 μm were obtained using a microtome.

**Electron microscopy.** Ultrastructural analysis was performed on artery samples fixed in 2.5% glutaraldehyde in 0.1 mmol/L cacodylate buffer (pH 7.4) at 4°C. Then fragments were post-fixed in 1% osmium tetroxide, dehydrated using graded alcohol series, and embedded in epoxy resin. Semi-fine sections (0.5 μm) were stained using toluidine blue. Ultrastructure sections (60 nm) were contrast-enhanced using uranyl acetate and lead citrate, and examined using a JEOL 1010 electron microscope (JEOL, Ltd., Tokyo, Japan) with a MegaView III camera (Olympus Soft Imaging Systems GmbH, Münster, Germany).

**Ultrasound.** Trans-thoracic ultrasound was performed in anesthesied mice (isoflurane 2.5%) using the Vevo 2100 Ultrasound system (Visual sonics Inc, Toronto, Ontario, Canada), equipped with an 18–38 MHz transducer. Mice were shaved at the thorax and abdomen and ultrasound gel is applied. Aorta diameters were observed and measured on longitudinal section. These measures have been carried out on 20-week-old male (n = 7 $Col3a1^{+/+}$ and n = 4 $Col3a1^{+/G182R}$) and female (n = 6 $Col3a1^{+/+}$ and n = 5 $Col3a1^{+/G182R}$) mice.

**RNA isolation and quantitative real-time PCR.** The descending aorta from 8-week-old males (n = 13 $Col3a1^{+/+}$ and n = 15 $Col3a1^{+/G182R}$) was excised after sacrificing the animal. The aorta was cleaned and the fat was removed then aorta was frozen in liquid nitrogen and stored at -80°C. Total RNA was extracted with the RNeasy micro kit (Qiagen, Valencia, CA), and the cDNA was synthesized with the iscript cDNA synthesis kit (Biorad, Hercules, CA). The SSoAdvanced SYBR Green supermix (Biorad) was used for real time PCR detection with the primers described in S1 Table. The reaction was run in a 3 technical replicates of RNA in a Biorad CFX96 with an annealing temperature of 59°C. The result was analyzed by the 2-ΔΔCt method normalized to *GAPDH* and *UBC*.

**Droplet digital PCR.** Total RNA were extracted using QIAzol Lysis Reagent (Qiagen, Hilden, Germany) from TA of $Col3a1^{+/G182R}$ mice and $Col3a1^{+/+}$ mice, stored at -80°C, according to the manufacturer's instructions. Total RNA were measured (NanoDrop One, Thermo Fisher, Waltham, MA), normalized to 10ng/μL and then reverse transcribed into cDNA using iScript cDNA Synthesis Kit (Bio-Rad, Hercules, CA). The cDNA was used as template for the droplet digital PCR (ddPCR) analysis as follows: ddPCR reaction mixture (12.5μL ddPCR Supermix for Probes (No dUTP) (Bio-Rad, Hercules, CA), 1 μL specific primers of WT and mutated alleles (S1 Table), 20 ng cDNA, and water up to 20 μL) was used for droplet generation. PrimePCR ddPCR Expression Probe Assay: Vim, Mouse (0.5X, Bio-Rad, Hercules, CA) was used as the internal control to normalize the expression of *Col3a1*. After droplet generation (QX100 Droplet Generator, Bio-Rad Laboratories, Hercules, CA), 40 μL of sample was manually transferred to a 96-well plate (Eppendorf, Hamburg, Germany). Amplification was performed (40 cycles of amplification, temperature of hybridization 61°C) and fluorescent intensity was measured in a QX100 Droplet Reader (Bio-Rad Laboratories, Hercules, CA) and the signal data was analysed using QuantaSoft Analysis Pro (1.0.596, Bio-Rad Laboratories, Hercules, CA).

**Immunofluorescence.** 5μm paraffin TA sections were deparaffinized, and antigens were unmasked using Citrate Buffer For Heat Induced Epitope Recovery, pH 6.0 (Clinisciences, Nanterre, France) at 95°C for 20 min. Membranes were permeabilised with 0.1% Triton-X100 in PBS for 10 min and blocked in 3% bovine serum albumin, 0.005% TBS-Tween in for 45 min. Then the sections were sequentially immunostained with primary antibody against collagen III (1:50, Arigo, Hsinchu, Taiwan) followed by Alexa594-conjugated donkey anti-goat IgG (Invitrogen, Carlsbad, CA); then primary antibody against collagen I (1:50, Novotec, Bron, France), followed by secondary antibody Alexa488-conjugated goat anti-rabbit IgG (Invitrogen, Carlsbad, CA). After DAPI and Sudan black staining, microscopy was performed with Axio imager ApoTome.2 (Zeiss, Oberkochen, Germany), images were acquired using ZenMicroscopy Software (Zeiss, Oberkochen, Germany) and images were recorded and treated

through NIS-Element AR Analysis Version 4.40 (Nikon, Tokyo, Japan). The overall intensities for collagens I and III were determined, 4–6 images were analyzed for each mouse, and the Collagen III/Collagen I ratio was calculated for $Col3a1^{+/+}$ and $Col3a1^{+/G182R}$ mice.

**Collagen quantification–Picrosirius red staining.** Collagen content of the aortic wall was assessed by staining sections of the descending TA from 24-week-old and untreated $Col3a1^{+/G182R}$ male mice and $Col3a1^{+/+}$ male mice with Picrosirius red staining. Digital images were acquired using the nanoZoomer HT scanner (Hamamatsu, Hamamatsu-city, Japan) with a magnification of 40X and a resolution of 0.23μm/pixel. Fiber (team 2, PARCC) was used to count the number of pixels corresponding to collagens and non-collagenous tissues after a learning process to assign the pixels to three categories: collagens, non-collagenous tissues and no tissue. Then the areas of aortas, intima-media and adventitia sections were evaluated with the total number of pixels and the relative amount of collagens was determined with the ratio between pixels of collagens in the entire aorta section or the intima-media and adventitia sections and the areas of corresponding sections.

**RNAseq.** After RNA extraction, RNA concentrations were obtained using nanodrop or a fluorometric Qubit RNA assay (Life Technologies, Grand Island, New York, USA). The quality of the RNA (RNA integrity number) was determined on the Agilent 2100 Bioanalyzer (Agilent Technologies, Palo Alto, CA, USA) as per the manufacturer's instructions.

To construct the libraries, 1 μg of high quality total RNA sample (RIN >8) was processed using TruSeq Stranded mRNA kit (Illumina) according to manufacturer instructions. Briefly, after purification of poly-A containing mRNA molecules, mRNA molecules are fragmented and reverse- transcribed using random primers. Replacement of dTTP by dUTP during the second strand synthesis will permit to achieve the strand specificity. Addition of a single A base to the cDNA is followed by ligation of Illumina adapters.

Libraries were quantified by qPCR using the KAPA Library Quantification Kit for Illumina Libraries (KapaBiosystems, Wilmington, MA) and library profiles were assessed using the DNA High Sensitivity LabChip kit on an Agilent Bioanalyzer. Libraries were sequenced on an Illumina Nextseq 500 instrument using 75 base-lengths read V2 chemistry in a paired-end mode.

After sequencing, a primary analysis based on AOZAN software (ENS, Paris) was applied to demultiplex and control the quality of the raw data (based of FastQC modules / version 0.11.5). Obtained fastq files were then aligned using STAR algorithm (version 2.5.2b) and quality control of the alignment realized with Picard tools (version 2.8.1). Reads were then count using Featurecount (version Rsubread 1.24.1) and the statistical analyses on the read counts were performed with the DESeq2 package version 1.14.1 to determine the proportion of differentially expressed genes between $Col3a1^{+/G182R}$ and $Col3a1^{+/+}$ (control) male mice. Statistically significant DEGs were selected by using the Limma moderated t test. Finally, the Benjamini & Hochberg procedure was for multiple testing adjustment to control the false discovery rate (FDR).

**Western blot.** Total proteins were extracted from TA of $Col3a1^{+/G182R}$ and $Col3a1^{+/+}$ 8 week-old male mice stored at -80˚C using a Protein Extraction Kit (Full Moon Biosystems, Sunnyvale, CA). Mini EDTA-free Protease Inhibitor Cocktail and PhosphoSTOP (Roche, Bâle, Switzerland) were added in the buffer of Protein Extraction Kit. Proteins were quantified using Pierce BCA Protein Assay Kit (Thermo Fisher, Waltham, MA). Separation was done on Mini-PROTEAN TGX Stain-Free Precast Gels (Bio-Rad, Hercules, CA), and proteins were transferred on Amersham Protran 0.2 NC membranes (GE Healthcare) for western blotting. Membranes were incubated in Ponceau S Solution (Sigma, Saint-louis, MO) to record picture by ChemiDoc XRS+ imaging System for normalization. After washes with Tris-buffered-saline supplemented with 0.05% Tween 20 (TBS-T), membranes were preblocked 1h in TBS-T/5%

BSA, then incubated overnight at 4˚C in TBS-T with 2% BSA with the antibodies against C-ter propeptide of collagen III (1:2500, # PAD195Mu01, Cloud-Clone Corp., Katy, TX), pERK (1:1000, #4370, Cell Signaling Technology, Danvers, MA), pPKC (1:1000, #ab75837, Abcam, Cambridge, UK), TGFβR1 (1:1000, #ab31013, Abcam, Cambridge, UK). After three washes in TBS-T, membranes were incubated 2h in TBS-T with 2% BSA containing secondary anti-rabbit antibodies-HRP conjugate (1:2000, #1721019 Bio-Rad, Hercules, CA). After three more washes in TBS-T, detection was carried out using SuperSignal West Pico PLUS Chemiluminescent Substrate (Thermo Fisher Scientific, Waltham, MA). Imaging was done using a ChemiDoc XRS+ imaging System (Bio-Rad, Hercules, CA). Quantification was performed using Image Lab Software Version 5.2.1.

**TGF-β pathway analysis.** TA were isolated and cells were lysed using Bio-Plex Cell Lysis Kit (Bio-Rad, Hercules, CA). The amount of proteins of each sample was measured with Pierce BCA protein Assay Kit (Thermo Fisher, Waltham, MA) and normalized to the smaller concentration obtened: 3.83μg. Then the 3 TGF-β isoforms: TGF-β1, TGF-β2 and TGF-β3, were determined using Bio-Plex Pro TGF-β 3-Plex Assay (171W4001M) (Bio-Rad, Hercules, CA).

**Drug treatment.** The drug trials were initiated at birth (in mother drinking water) to 24 weeks of age for propranolol, losartan, amlodipine, and the association propranolol-amlodipine, whereas they were initiated in 4-week-old $Col3a1^{+/G182R}$ mice and continued until 24 weeks of age for losartan (both conditions), celiprolol, hydralazine and the association celiprolol-hydralazine. The trials were performed in $Col3a1^{+/G182R}$ male offsprings since they had the higher spontaneous rate of aortic rupture. Propranolol (Sigma-Aldrich, St. Louis, MO) was dissolved in drinking water to give a concentration of 0.5 g/L and an estimated dose of 115 mg/kg/day. Celiprolol (Acer Therapeutics, The Woodlands, TX) was dissolved in drinking water to give a concentration of 1.28 g/L and an estimated dose of 250 mg/kg/day. Losartan (Sigma-Aldrich, St. Louis, MO) was dissolved in drinking water to give a concentration of 0.6 g/L and an estimated dose of 135 mg/kg/day. Amlodipine (Sigma-Aldrich, St. Louis, MO) was dissolved in drinking water with 1% of ethanol to give a concentration of 0.033 g/L and an estimated dose of 7.5 mg/kg/day. Hydralazine (Sigma-Aldrich, St. Louis, MO) was dissolved in drinking water to give a concentration of 0.25 g/L and an estimated dose of 45 mg/kg/day. The associations of Celiprolol-Hydralazine and Amlodipine-propranolol were given with the same dose as monotherapy.

Another drug trial with losartan was initiated in pregnant mice, continued in the mothers until the weaning and then in the offsprings until 24 weeks of age. Thereafter, a washout period of 24 additional weeks was established until 48 weeks of age.

**Plasma Celiprolol concentration.** We used an ultra-high performance liquid chromatography coupled with tandem mass spectrometry (UHPLC-MS/MS) assay to quantify celiprolol in mice plasma samples that were stored at –20˚ C before analysis.

For the analysis, samples were thawed and centrifugated. A 25 μL aliquot of the sample was diluted by 1/4 in a solution of water/acetonitrile (90/10, vol/vol). The diluted sample (100 μL) was then prepared by precipitation with 600 μL of acetonitrile/formic acid (99/1, vol/vol). After centrifugation (10 000 rpm for 8 minutes), 600 μL of the supernatant was evaporated to dryness with nitrogen and reconstituted with 200 μL of water/acetonitrile (90/10, vol/vol). After homogenization and centrifugation (10 000 rpm for 8 minutes), the extracted solution was analyzed by UHPLC–MS/MS. The system used was an Acquity UPLC hyphenated to a Xevo TQS triple quadrupole mass spectrometer (Waters Corp., Milford, Massachusetts, USA) fitted with an electrospray ionization source and running in positive ionization mode. A Waters Acquity UPLC BEH C18 (1.8 μm, 2.1x50mm) column was used for the chromatographic separation, with temperature maintained at 40˚C.

The mobile phase was delivered at 0.4 mL/min, in gradient mode, with 0.1% acetic acid in water as mobile phase (A) and acetonitrile as mobile phase (B). The elution program increased from 10% to 65% phase (B) in 3 minutes, and from 65% to 90% phase (B) in 30 seconds. These conditions were maintained for 30 seconds before return to initial conditions and reequilibration for 1 minute.

Capillary voltage was set at 2 kV and cone voltage was set at 45V, 20V and 20V for celiprolol, hydralazine and the internal standard d5-bumetanide respectively. The source temperature was maintained at 150˚C. The desolvatation gas flow was set to 800 L/h and the cone gas flow to 150 L/h.

The mass spectrometer was set to targeted multiple reaction monitoring mode, and used the following mass transition (m/z): 380.2 > 251.1 for celiprolol, 161.1 > 89.1 for hydralazine and 369.9 > 244.0 for the internal standard d5-bumetanide. The Mass-Lynx software (V4.1 SCN805) was used for data acquisition.

## Statistical analysis

All data are expressed as mean ± SEM.

Kaplan-Meyer survival analyses and Student t-tests were carried out using GraphPad Prism version X7.0a for Macintosh (GraphPad Software).

We performed Kaplan-Meier survival analyses to draw and compare survival rates between the two genotypes (*Col3a1*$^{+/+}$ and *Col3a1*$^{+/G182R}$), sexes (*Col3a1*$^{+/G182R}$ males and females), for Ang II infusion (*Col3a1*$^{+/+}$ and *Col3a1*$^{+/G182R}$) and for each drug treatment (*Col3a1*$^{+/G182R}$ with or without drug). A Logrank (Mantel-Cox) test was used to calculate the statistical significance between two curves for all survival studies.

In order to check we could correctly interpret the effect of the different drugs on the survival in our model, we showed there was no statistical difference of survival rate between the non-treated mice used as controls in the different treatment groups (S10 Fig).

When the Shapiro-Wilk normality test was passed, a student t-test for unpaired series was used to compare between *Col3a1*$^{+/G182R}$ and *Col3a1*$^{+/+}$ mice: the area and collagen density in the different aorta sections, the expression of total col3a1 mRNA, the quantification of western blot and qPCR results and the quantification of TGF-β isoforms. For the relative expression of collagen III by immunofluorescence that did not pass Shapiro-Wilk normality test, Wilcoxon (nonparametric) test was performed to evaluate significance between groups.

A comparative test between theoretical and observed proportions was performed to compare the specific allele expression within each group of mice.

To compare SBP and HR between treatments and water (except amlodipine-propranolol which was compared to amlodipine) over the 24-week period, we used generalized linear mixed-effects model (GLMM) when it was valid, using RStudio version 1.3.1056; 2009–2020 (RStudio, PBC.) The validity of the GLMM was assessed using Rainbow test (to check the fit of the regression model), Durbin-Watson test (to verify if the residuals from a linear model are correlated), Shapiro-Wilk test (to check the normality of residuals) and Breush-Pagan test (to check the homoscedasticity of residuals). Prior to generalized linear mixed model (GLMM), all measures were tested for repeated measures ANOVA (RMA) and a Bonferroni correction was applied, leading to the same results as GLMM.

Statistical significance was deemed to have been achieved when $p < 0.05$.

## Supporting information

**S1 Fig. Gene targeting strategy and ES screen for Col3a1 knock-in mice generation. A.** Gene targeting strategy used for generating the *Col3a1*$^{+/G182R}$ mutated allele. The WT and

targeted alleles are shown before and after FLP recombination (with and without the pgk-neorcassette respectively). Boxes represent exons (the number of exon is indicated) and those with pgk-neor represent the neomycin selection cassette. Black boxes correspond to FRT recombination sites, the asterisk represents the mutated site in exon 6. For Southern blot analysis, the 5' probe used is shown as a red rectangle upper the Col3a1 WT and targeted allele and relevant restrictions sites are shown (BamH1). For genotype analysis, the col3a1 reverse (green arrows) and forward (red arrows) primers are shown on both WT and mutated allele. **B.** The G to A substitution is checked in ES cell clones by Sanger sequencing. Compared with WT ES cell clones, heterozygous ES cell clones reveal the substitution c.547G>A leading to an amino-acid change p.G182R. The asterisk represents the mutated nucleotide on exon 6. **C.** Genotype analysis of WT and Col3a1 mutated mice. Both WT and mutated alleles were detected by PCR using Col3a1 primers.
(TIFF)

**S2 Fig. Survival rate of *Col3a1^{+/G182R}* female mice in weaning period.** Kaplan-Meier Survival curve of *Col3a1^{+/G182R}* female mice (n = 11) in three successive weaning periods (21 days for each period). Six, two and one deaths are observed during the first, second and third weaning periods respectively. Thoracic aortic rupture is observed at autopsy for 90.9% (n = 10/11) of mice.
(TIFF)

**S3 Fig. In vivo aorta diameter measured in 6-week-old *Col3a1^{+/+}* and *Col3a1^{+/G182R}* mice by ultrasound. A**. Ascending and descending thoracic aorta in *Col3a1^{+/G182R}*. White arrows indicate the ascending and descending thoracic aortic diameter. **B.** Abdominal aorta in *Col3a1^{+/G182R}*. Blue arrow indicates the abdominal aortic diameter. **C-D.** Measurement of the three diameters in males and females respectively. Error bars show mean ± SEM. No significant differences were found using Student t-test when the three diameters were compared between Col3a1^{+/G182R} and Col3a1^{+/+} mice in both sexes (p>0.05).
(TIFF)

**S4 Fig. Heterogeneity of collagen fibers in *Col3a1^{+/G182R}* mice.** The proportion (%) of collagen fibrils with a given diameter (nm) shows heterogeneity in the distribution for *Col3a1^{+/G182R}* mice compared to *Col3a1^{+/+}* controls in the aorta. In the table, the diameters of the collagen fibrils show a wider range in *Col3a1^{+/G182R}* mice than in *Col3a1^{+/+}* mice. The % fibers represent the number of collagen fibrils per unit area considering the number of collagen fibrils in *Col3a1^{+/+}* mice as the reference. The % fibers reveal a lower density of collagen fibrils in *Col3a1^{+/G182R}* mice than in *Col3a1^{+/+}* mice.
(TIFF)

**S5 Fig. Relative mRNA expression of HSP47, ATF6, ATG5 and ATG7 in thoracic aorta of *Col3a1^{+/G182R}* and *Col3a1^{+/+}* mice determined by q-PCR.** The expression of HSP47, ATF6, ATG5 and ATG7 is significantly different in the thoracic aorta of Col3a1^{+/G182R} mice compared to *Col3a1^{+/+}* controls (Student t-test, p<0.05).
(TIFF)

**S6 Fig. Immunofluorescence of descending TA sections in *Col3a1^{+/G182R}* mice. A.** Immunofluorescent staining of descending TA sections of *Col3a1^{+/+}* (n = 10) and *Col3a1^{+/G182R}* (n = 10) mice to show the distribution of collagens I and III. The sections are stained with collagen I (green) and collagen III (red) polyclonal antibodies and the cell nuclei were stained with DAPI (blue). The anti-collagen III antibody failed to recognize mature collagen III in *Col3a1^{+/G182R}* mice. **B.** Relative quantification of the mature collagen III using the Collagen

III/Collagen I immunofluorescence intensity ratio: the quantity of collagen III was collapsed probably due to conformational changes which lead to the absence of detection of collagen III, in the TA of *Col3a1*$^{+/G182R}$ mice compared to *Col3a1*$^{+/+}$ controls (Wilcoxon test, p<10$^{-4}$). (TIFF)

**S7 Fig. Plasma celiprolol concentration measurement.** Plasma concentration of celiprolol in *Col3a1*$^{+/G182R}$ (n = 80) mice receiving celiprolol or the association celiprolol-hydralazine and in *Col3a1*$^{+/G182R}$ (n = 26) mice receiving water, all of them being included in the celiprolol and celiprolol-hydralazine protocols. The celiprolol concentration was not significantly difference between the two treated groups. Data are expressed as the mean ± SEM. (TIFF)

**S8 Fig. Consequences of treatment by amlodipine and the association amlodipine–propranolol in *Col3a1*$^{+/G182R}$ female mice. A.** Survival comparison between Amlodipine and ethanol 1%, and between the association amlodipine-propranolol and amlodipine (monotherapy). Kaplan-Meier Survival curve (red curve) for comparing *Col3a1*$^{+/G182R}$ treated with amlodipine (n = 12) to *Col3a1*$^{+/G182R}$ treated with ethanol 1% (n = 33). Amlodipine worsens the mortality despite no significant difference is observed using Log-Rank (Mantel-Cox) analysis (p = 0.1363). Kaplan-Meier Survival curve (brown curve) for comparing *Col3a1*$^{+/G182R}$ treated with amlodipine-propranolol (n = 10) to *Col3a1*$^{+/G182R}$ treated with amlodipine (n = 12). The association improves slightly the survival despite insignificant difference is calculated using Log-Rank (Mantel-Cox) analysis (p = 0.1365). **B.** SBP and HR comparison between amlodipine and ethanol 1%, and between the association amlodipine-propranolol and amlodipine (monotherapy). A significant decrease in SBP (linear mixed-effects model, p = 8.17×10–3) was observed that was associated with no change in HR (student t-test, p>0.05 for each time of the 24-week follow-up period) when comparing amlodipine to ethanol 1% (red bars and curve). (TIFF)

**S9 Fig. Normal BP in *Col3a1*$^{+/G182R}$ mice and decrease on Losartan. A.** Day SBP before (4 days = basal) or during oral administration (4 days) of losartan (135 mg/kg/day) in *Col3a1*$^{+/G182R}$ mice (n = 8) compared to *Col3a1*$^{+/+}$ mice (n = 7) examined with a telemetric system. **B.** Night SBP before (4 days = basal) or during oral administration (4 days) of losartan (135 mg/kg/day) in the same *Col3a1*$^{+/G182R}$ mice (n = 8) and *Col3a1*$^{+/+}$ mice (n = 7). **C.** Comparison of the mean of the day SBP before and during administration of losartan in the same *Col3a1*$^{+/G182R}$ mice (n = 8) and *Col3a1*$^{+/+}$ mice (n = 7). Significant decrease is observed in both groups. **D.** Comparison of the mean of the night SBP before and during administration of losartan in the same *Col3a1*$^{+/G182R}$ mice (n = 8) and *Col3a1*$^{+/+}$ mice (n = 7). Significant decrease is observed in both groups. (TIFF)

**S10 Fig. Independent groups of *Col3a1*$^{+/G182R}$ control mice performed over time.** Kaplan-Meier Survival curve for comparing the 5 *Col3a1*$^{+/G182R}$ male control groups. No significant difference is observed using Log-Rank (Mantel-Cox) analysis (p = 0.7881). (TIFF)

**S1 Table. Primers used for the generation of the knock-in model, the genotyping of *Col3a1*$^{+/G182R}$ and *Col3a1*$^{+/+}$ mice, the droplet PCR and the RT-qPCR.** (XLSX)

**S2 Table. Basic parameters (weight, SBP and HR) in *Col3a1*$^{+/G182R}$ and *Col3a1*$^{+/+}$ mice throughout the 24-week follow-up period.** Data are expressed as the mean ± SEM. The

number of mice at each time are indicated above the mean ± SEM. Student-t test.
(XLSX)

**S3 Table. Differential expression of the 223 genes belonging to the MAPK signaling pathway of KEGG database (corresponding to the PLC/IP3/PKC/ERK signaling pathway) in the transcriptomic data of *Col3a1*$^{+/G182R}$ mice.** Significant differential expressed genes are mentioned in bold. Genes not found in transcriptomic data are highlighted in grey. Adjusted p-value with the Benjamini-Hochberg method.
(XLSX)

**S4 Table. Expression of genes of the PLC/IP3/PKC/ERK signaling pathway in the transcriptomic data of *Col3a1*$^{+/G182R}$.** Significant differential expressed genes are mentioned in bold. Genes not found in transcriptomic data are highlighted in grey. Adjusted p-value with the Benjamini-Hochberg method.
(XLSX)

**S5 Table. Quantification of the 3 TGF-β isoforms in *Col3a1*$^{+/G182R}$ mice.** Amount of the 3 TGF-β isoforms per μg of protein extracted from TA. Data are expressed as the mean ± SEM. Student t-test.
(XLSX)

**S6 Table. Comparison of therapeutic strategies (monotherapies) with measurement of SBP and HR in *Col3a1*$^{+/G182R}$ male mice.** Data are expressed as the mean ± SEM. The number of mice at each time are indicated above the mean ± SEM. Student-t test or Log-Rank test (Mantel-Cox) for the survival.
(XLSX)

**S7 Table. Comparison of therapeutic strategies (dual therapies in male mice and Amlodipine therapies in female mice) with measurement of SBP and HR in *Col3a1*$^{+/G182R}$ male mice.** Data are expressed as the mean ± SEM. The number of mice at each time are indicated above the mean ± SEM. Student-t test or Log-Rank test (Mantel-Cox) for the survival.
(XLSX)

**S8 Table. Differential expression of the 6 relevant genes belonging to the Ang II signaling pathway in the transcriptomic data of *Col3a1*$^{+/G182R}$ mice.** Adjusted p-value with Benjamini-Hochberg method.
(XLSX)

**S9 Table. Numerical data of the study.**
(XLSX)

## Acknowledgments

The authors thank Eric Clauser for his critical revision of the manuscript, Hani Nematalla, Emmanuelle Plaisier, Marie Crahes, Patrick Bruneval and Emmanuel Messas for their participation to the study.

The authors would like to thank Nicolas Perez, Corina Suldac, Guillaume Gueguin and Angélique Fournet from the Plateforme ERI of the PARCC, for the care of animals and good advice. They would also like to thank Corinne Lesafre for her help in the histological analyses (Histology plateform, PARCC), Daniel Balvay for his advices and teaching to use Fiber v5.2 (team 2 In vivo Imaging Research, PARCC), the team of the Genom'IC platform of the Cochin Institute, especially Benjamin Saintpierre and Franck Letourneur for the quality or their services, Morgane Le Gall for her teaching to use GSEA (Proteomic Plateforme 3P5 of the Cochin

Institute), Benjamin Kably and Christophe Simian for perfection and performing plasma assays of celiprolol.

## Author Contributions

**Conceptualization:** Juliette Hadchouel, Xavier Jeunemaitre.

**Data curation:** Anne Legrand, Charline Guery, Xavier Jeunemaitre.

**Formal analysis:** Anne Legrand, Charline Guery, Julie Faugeroux, Erika Fontaine, Carole Beugnon, Amélie Gianfermi, Irmine Loisel-Ferreira, Marie-Christine Verpont, Salma Adham.

**Funding acquisition:** Xavier Jeunemaitre.

**Investigation:** Anne Legrand, Charline Guery, Julie Faugeroux, Erika Fontaine, Carole Beugnon, Amélie Gianfermi, Irmine Loisel-Ferreira, Marie-Christine Verpont, Salma Adham.

**Methodology:** Juliette Hadchouel, Xavier Jeunemaitre.

**Project administration:** Tristan Mirault, Juliette Hadchouel, Xavier Jeunemaitre.

**Software:** Anne Legrand, Charline Guery, Salma Adham.

**Supervision:** Tristan Mirault, Juliette Hadchouel, Xavier Jeunemaitre.

**Validation:** Anne Legrand, Charline Guery, Tristan Mirault.

**Writing – original draft:** Anne Legrand, Charline Guery, Erika Fontaine.

**Writing – review & editing:** Tristan Mirault, Juliette Hadchouel, Xavier Jeunemaitre.

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
