## [Decision Letter · Decision Letter 0]

30 Nov 2021

Dear Dr Legrand,

Thank you very much for submitting your Research Article entitled 'Comparative therapeutic strategies for preventing aortic rupture in a mouse model of vascular Ehlers Danlos syndrome' to PLOS Genetics.

The manuscript was fully evaluated at the editorial level and by independent peer reviewers. The reviewers appreciated the attention to an important topic but identified some concerns that we ask you address in a revised manuscript

We therefore ask you to modify the manuscript according to the review recommendations. Your revisions should address the specific points made by each reviewer.

[LINK]

Yours sincerely,

Fransiska Malfait

Guest Editor

PLOS Genetics

Scott Williams

Section Editor: Natural Variation

PLOS Genetics

Reviewer's Responses to Questions

**Comments to the Authors:**

Reviewer #1: This investigation describes the phenotype of a newly created transgenic mouse line with respect to mortality, biochemical pathways, and changes in survival in response to several pharmacological manipulations. This mouse line was created to recapitulate characteristics of the vascular Ehler-Danlos syndrome observed in humans by modifying the collagen III product. In this regard, there are published reports from at least two other transgenic mouse lines that use different modifications of the collagen III gene. The authors show that their transgenic model are similar to the human condition in many important ways (as do the other studies) but does differ with respect to survival benefits in response to celiprolol, which has β1-antagonistic and β2-agonistic properties. In addition, the authors highlight some important differences in activation of certain biochemical pathways between this transgenic line and those previously published.

The experiments have been carefully thought out and the number of animals in experimental cohorts are, in general, adequate. Statistical analyses are appropriate for this set of studies – with the possible exception that changes in systolic blood pressure and heart rate may have benefitted from applying repeated measures ANOVA tests given that the same animals were tracked over time. This may have been done as part of the mixed-effects modeling; but, if that is the case, it is not explicitly stated. In addition, comparisons between groups should be corrected for the effects of multiple comparisons (Tukey’s, Bonferroni’s, or Scheffe’s tests).

The display analysis of the survival analysis for the lactating mothers (Figure S2) seems a little strange to be put on a continuous time scale – unless, the mothers that were alive through the pre-weaning period from the previous littler were immediately pregnant. That would suggest that these mothers may have been pregnant and lactating at the same time. If there was a gap between conceptions, it may be more fruitful to overlay the survival curves and then indicate the “number at risk” through 21 days for each of the three lactation periods.

Finally, were the data for mRNA expression and immunoblotting checked for normality of distribution? If not normally distributed, please consider log transforming the data prior to running parametric tests.

Minor comments:

Introduction: The sentence beginning on line 64 (“By far the most…”) is difficult to follow. Please consider rephrasing.

Introduction: On line 77, it may be better to state “thinner arterial walls” rather than “thinner arterial thickness”.

Introduction: On line 83, “arterial” may be deleted.

Results: Line 256: “Supplemental Figure 7” should be “Supplemental Figure 8”.

Discussion: all through: “coherence” may be better stated as “concordance” or “consistent with”.

Discussion: Line 379: “closed” should be “close”.

Methods: Line 409: Please remove the opening parenthesis before “This”.

Methods: Line 512 – “Echography” should be “ultrasound” – and also through the paragraph. If you performed any echocardiography, you have not shown the results.

Methods: Echography paragraph – please change to past tense to be consistent with the rest of the methods section.

Methods: Line 515 – “transducer” is missing after “MHz”.

Reviewer #2: Even though your research was extensive and broad in scope, I found it easy to read your manuscript and follow the work that you did. I think it should be published with haste as it will be useful for planning clinical trials in vEDS. There are just a couple improvements that would make the manuscript easier to read.

1. Confusing numbers Aortic samples were taken after sacrifice at 4 weeks, 6 weeks, 24 weeks, each time studying small sample numbers. At times the n (number individual aorta samples) is not written in text but only in figure. I find it helpful to have the n= number near the p value. Please provide the n number in the text itself.

2. Please also clarify or confirm that the samples taken for aortic study were from untreated animals, both wild type and heterozygous COL3A1 mutants.

2. COL3A1 protein expression: Is the type III collagen antibody used in immunoflorescence and in Western blot from the p-ter region of the molecule? I found a description of immunofluorescent antibody location only. Might this contribute to the discrepant results (increased proIII on Western and absent on immunofluorescence).

3. Please include the p. number for the G182R in the mouse model to clarify that it is not a triple helical number.

**Have all data underlying the figures and results presented in the manuscript been provided?**

Reviewer #1: Yes

Reviewer #2: Yes

PLOS authors have the option to publish the peer review history of their article (what does this mean?). If published, this will include your full peer review and any attached files.

Reviewer #1: No

Reviewer #2: No

---

## [Decision Letter · Decision Letter 1]

28 Jan 2022

Dear Dr Legrand,

We are pleased to inform you that your manuscript entitled "Comparative therapeutic strategies for preventing aortic rupture in a mouse model of vascular Ehlers Danlos syndrome" has been editorially accepted for publication in PLOS Genetics. Congratulations!

Yours sincerely,

Fransiska Malfait

Guest Editor

PLOS Genetics

Scott Williams

Section Editor: Natural Variation

PLOS Genetics

Comments from the reviewers (if applicable):

Reviewer's Responses to Questions

**Comments to the Authors:**

Reviewer #1: Thank you for responding to my comments. I have just a very minor comment.

What do you mean by "degreased" on line 495?

Reviewer #2: Thank you for revisions.

**Have all data underlying the figures and results presented in the manuscript been provided?**

Reviewer #1: None

Reviewer #2: Yes

PLOS authors have the option to publish the peer review history of their article (what does this mean?). If published, this will include your full peer review and any attached files.

Reviewer #1: No

Reviewer #2: No

**Data Deposition**

http://datadryad.org/submit?journalID=pgenetics&manu=PGENETICS-D-21-01168R1

**Press Queries**

---

## [Editor Report · Acceptance letter]

23 Feb 2022

PGENETICS-D-21-01168R1 

Comparative therapeutic strategies for preventing aortic rupture in a mouse model of vascular Ehlers-Danlos syndrome 

Dear Dr Legrand, 

We are pleased to inform you that your manuscript entitled "Comparative therapeutic strategies for preventing aortic rupture in a mouse model of vascular Ehlers-Danlos syndrome" has been formally accepted for publication in PLOS Genetics! Your manuscript is now with our production department and you will be notified of the publication date in due course.

With kind regards,

Zsofia Freund

PLOS Genetics

On behalf of:
